# A Theory for Token-Level Harmonization in Retrieval-Augmented Generation

**Shicheng Xu**[1,2], **Liang Pang**[1]*, **Huawei Shen**[1], **Xueqi Cheng**[1]

[1]CAS Key Laboratory of AI Safety, Institute of Computing Technology, Chinese Academy of Sciences
[2]University of Chinese Academy of Sciences
`{xushicheng21s,pangliang,shenhuawei,cxq}@ict.ac.cn`

## Abstract

Retrieval-augmented generation (RAG) utilizes retrieved texts to enhance large language models (LLMs). Studies show that while RAG provides valuable external information (benefit), it may also mislead LLMs (detriment) with noisy or incorrect retrieved texts. Although many existing methods attempt to preserve benefit and avoid detriment, they lack a theoretical explanation for RAG. The benefit and detriment in the next token prediction of RAG remain a 'black box' that cannot be quantified or compared in an explainable manner, so existing methods are data-driven, need additional utility evaluators or post-hoc. This paper takes the first step towards providing a theory to explain and trade off the benefit and detriment in RAG. First, we model RAG as the fusion between distribution of LLM's knowledge and distribution of retrieved texts. Then, we formalize the trade-off between the value of external knowledge (benefit) and its potential risk of misleading LLMs (detriment) in next token prediction of RAG by distribution difference in this fusion. Finally, we prove that the actual effect of RAG on the token, which is the comparison between benefit and detriment, can be predicted without any training or accessing the utility of retrieval. Based on our theory, we propose a practical novel method, **Tok-RAG**, which achieves collaborative generation between the pure LLM and RAG at token level to preserve benefit and avoid detriment. Experiments in real-world tasks using LLMs such as OPT, LLaMA-2, and Mistral show the effectiveness of our method and support our theoretical findings. Code is available[1].

## 1 Introduction

Retrieval-augmented generation (RAG) has shown promising performance in enhancing Large Language Models (LLMs) by integrating retrieved texts (Xu et al., 2023; Shi et al., 2023; Asai et al., 2023; Ram et al., 2023). Studies indicate that while RAG provides LLMs with valuable additional knowledge (**benefit**), it also poses a risk of misleading them (**detriment**) due to noisy or incorrect retrieved texts (Ram et al., 2023; Xu et al., 2024b;a; Jin et al., 2024; Xie et al., 2023). Existing methods attempt to preserve benefit and avoid detriment by adding utility evaluators for retrieval, prompt engineering, or fine-tuning LLMs (Asai et al., 2023; Ding et al., 2024; Xu et al., 2024b; Yoran et al., 2024; Ren et al., 2023; Mallen et al., 2022; Jiang et al., 2023). However, existing methods are data-driven, need evaluator for utility of retrieved texts or post-hoc. A theory-based method, focusing on core principles of RAG is urgently needed, which is crucial for reliable improvements without relying on additional training or utility evaluators and improving our understanding for RAG.

This paper takes the first step in providing a theoretical framework to explain and trade off the benefit and detriment at token level in RAG and proposes a novel method to preserve benefit and avoid detriment based on our theoretical findings. Specifically, this paper pioneers in modeling next token prediction in RAG as the fusion between the distribution of LLM's knowledge and the distribution of retrieved texts as shown in Figure 1. Our theoretical derivation based on this formalizes the core of this fusion as the **subtraction** between two terms measured by the distribution difference: one is **distribution completion** and the other is **distribution contradiction**. Further analysis indicates that

---

*Corresponding Author
[1]`https://github.com/xsc1234/Tok-RAG`

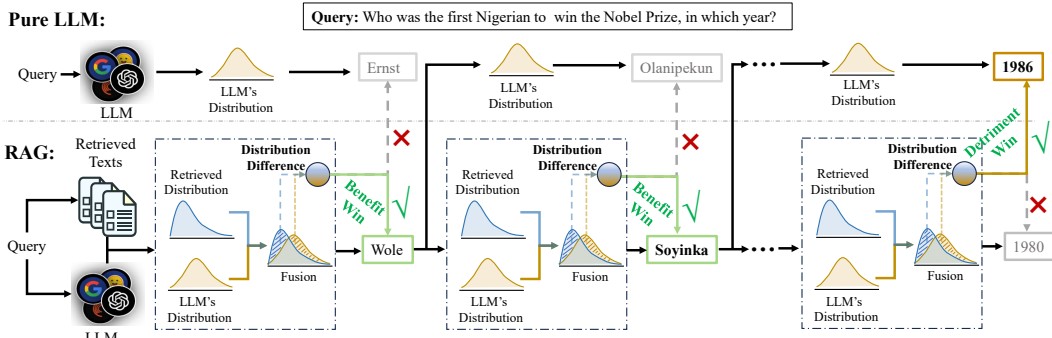

Figure 1: Framework of our Tok-RAG. It performs collaborative generation between pure LLM and RAG at the token-level by comparing benefit and detriment based on our theoretical findings about distribution difference. The selected tokens at each step are used as the prefix for both pure LLM and RAG. Tok-RAG preserves benefit and avoids detriment without any training or utility evaluators.

the distribution completion measures how much out-of-distribution knowledge that retrieved texts provide to LLM in the prediction of the next token, representing the **benefit** of RAG for LLM at the token level. However, since the retrieved texts may contain noisy or incorrect information, posing the risk of misleading the LLM, distribution contradiction captures this risk by measuring the degree of conflict between the LLM's knowledge and the external knowledge from the retrieved texts. This represents the **detriment** of RAG for LLM at token level. Thus, we prove that the fusion between distribution of LLM's knowledge and the retrieved texts in next token prediction of RAG is governed by the combined effect of benefit and detriment, whose relationship is described as subtraction.

In this way, we successfully decouple benefit and detriment from next token prediction of RAG and formalize them as the subtraction between two terms. This subtraction describes the trade-off between the value of external knowledge and its potential risk of misleading LLM without accessing the utility of the retrieved texts. We then prove that value of this subtraction is approximately positively correlated with the similarity between the representation of RAG's output and the representation of retrieval texts. Finally, we establish a theory to predict the actual effect of RAG at token level. Specifically, in next token prediction given prefix and retrieved texts, this theory determines whether the benefit brought by retrieved texts on the prediction of the token outweighs the detriment.

Based on our theoretical results, we propose a practical novel method called **Tok-RAG** that can achieve collaborative generation between pure LLM and RAG at token level to preserve benefit and avoid detriment **without any training or additional modules for utility evaluation of retrieved texts**. As shown in Figure 1, pure LLM and RAG generate the texts in parallel. At the generation step where LLM and RAG generate the different tokens, **Tok-RAG** uses our theoretical results to determine which token will be selected by comparing the values of benefit and detriment brought by RAG to the token. Experimental results in real-world tasks such as Q&A and Long-Form Q&A based on LLMs including OPT, LLaMA-2, and Mistral show the effectiveness of our method and support our theoretical results. Our method does not need any additional modules or training but outperforms baselines that need additional modules and fine-tuning LLMs by just accessing to the layers and logits in inference, which indicates that our theoretical results are essential and fundamental for RAG. The main contributions of this paper are:

• This paper takes the first step in theoretically giving the essential explanation of benefit and detriment in RAG to make them explainable and comparable at token-level.

• We provide a theory that actual effect of RAG (i.e., the comparison between benefit and detriment) can be predicted at token level by representation similarity without any training or accessing the utility of the retrieved texts, which is significant for fine-grained preserving benefit and avoiding detriment in practical applications of RAG.

• Based on the theoretical results, we propose a practical novel method to enable pure LLM and RAG to collaboratively generate at token level. Experimental results on real-world tasks across different LLMs show the effectiveness of our method and support our theoretical results.

Figure 2: Derivation path of our theory. Reference: Equation 9, Theorem 1, 2, 3 and Corollary 1, 2, 3.

## 2 THEORY TO UNDERSTAND BENEFIT AND DETRIMENT AT TOKEN LEVEL

Although retrieved texts can provide LLM with external knowledge (benefit), they also pose a risk of misleading LLM due to the noise within the retrieved content (detriment). This section provides the theoretical framework to explain and trade off benefit and detriment in RAG at token level. First, we propose to use latent variable model to model the next token prediction in RAG as the fusion between the distribution of LLM's knowledge and the distribution of retrieved texts. Second, our theoretical derivation formalizes the core of this fusion as the subtraction between two terms measured by the distribution difference: one is **distribution completion**, and the other is **distribution contradiction**. Our further discussion shows that distribution completion is actually the benefit and distribution contradiction is detriment in the next token prediction of RAG. Last but not least, we provide a theory to predict the actual effect of RAG at token level. Figure 2 shows the derivation path of our theory.

### 2.1 PRIMARY DEFINITIONS AND ASSUMPTIONS

**Retrieved Texts List $R$.** $R = \{r_1, r_2, r_3, ..., r_n\}$ is the list that contains retrieved passages for RAG, in which $r_i$ is a passage. In textual format of $R$, $r_{i-1}$ and $r_i$ is separated by a delimiter such as "[Retrieved Passage]", denoted as $d$.

**The Distribution of Retrieved Texts List.** $p_R(\cdot)$ is the distribution of retrieved texts list $R$. It is the language modeling distribution just like the distribution $p(\cdot)$ of the LLM. This distribution $p(\cdot)$ is learned through training LLM on large corpus with next-token prediction paradigm. $p(x_i|x_{1:i-1})$ represents the probability distribution predicted by the LLM for $x_i$, given the prefix $x_{1:i-1}$. The LLM is capable of generating reasonable and coherent text based on the distribution patterns of vocabulary and context learned from the large training data. Here, $p_R(\cdot)$ refers to the distribution $p(\cdot)$ learned when the training data that is limited to the retrieved passages list $R$, and it represents the vocabulary and contextual distribution patterns of the retrieved passages list $R$. Define $r_i$ is the $i$-$th$ passage in the retrieved passages list $R$, so $p_R(r_i)$ represents the joint probability of the natural language sequence $r_i = [r_i^1, r_i^2, r_i^3, ..., r_i^l]$ under the distribution $p_R(\cdot)$, in which $r_i^x$ is the $x$-$th$ token in passage $r_i$. Specifically, according to the chain rule, $p_R(r_i)$ can be decomposed into the product of a series of conditional probabilities:

$$p_R(r_i) = p_R(r_i^1) \cdot p_R(r_i^2|r_i^1) \cdot p_R(r_i^3|r_i^1, r_i^2) \cdots p_R(r_i^l|r_i^1, r_i^2, r_i^3, ..., r_i^{l-1}),$$

**Latent Variable Inference as Hidden Markov Model.** Inspired by previous studies that prove LLMs implicitly perform latent variable inference (Zhang et al., 2023; Wang et al., 2024), we first propose to analyze RAG by latent variable inference like Hidden Markov Model (HMM) Xie et al. (2021), where the latent concept $z$ determines the transition probability matrix between hidden states $h$, and both hidden states and latent concepts together determine the prediction of the token.

**Assumptions.** Based on previous work (Xie et al., 2021; Zhang et al., 2023), we make the following assumptions that:

**Assumption 1.** *All tokens can be predicted, which means that for every token $x$, there is some hidden state $h$ lower bounds it that $p(x|h, z^*) > c_1 > 0$.*

**Assumption 2.** *Delimiter $d$ is an important distinguishing signal between each passage $r$ in the retrieved texts $R$. The delimiter hidden state $h^d$ implies that $p(d \mid h^d, z) = 1$, meaning that the hidden state $h^d$ uniquely determines the output $d$ when $z$ is given, making the probability of observing $d$ equal to 1. For any delimiter hidden state $h^d$ and other hidden state $h$, there are upper and lower bounds on the transition probability from $h$ to $h^d$: $0 \le c_2 \le p(h^d|h, z) \le c_3$.*

## 2.2 Model Next Token Prediction in RAG as Distribution Fusion

We propose to analyze RAG by latent variable inference to model the next token prediction in RAG as the fusion between the distribution of LLM's knowledge and the distribution of retrieved texts. Given the prefix $x_{1:i-1} = \{x_1, x_2, ...x_{i-1}\}$, from the perspective of the latent variable inference, the probability distribution of the token $x_i$ at the $i$-th step is described as this:

$$p(x_i|x_{1:i-1}) = \int_{\mathcal{Z}} p(x_i|x_{1:i-1}, z)p(z|x_{1:i-1})\,dz, \tag{1}$$

in which $\mathcal{Z}$ is the space of high dimensional concept variable, $p(z|x_{1:i-1})$ is the probability that the model samples latent concept $z$ from $\mathcal{Z}$ given the prefix $x_{1:i-1}$, and $p(x_i|x_{1:i-1}, z)$ means the probability for token $x_i$ conditioned on the prefix $x_{1:i-1}$ and the sampled latent concept $z$. $p(x_i|x_{1:i-1})$ can be obtained by integrating over all latent concepts from the space $\mathcal{Z}$. Latent variable model has been applied in many methods such as LDA (Blei et al., 2003). Recent studies prove that in-context learning of LLMs can also be seen as the latent variable model, in which the LLMs sample the concept across the input examples (Xie et al., 2021; Zhang et al., 2023). Inspired by this, we analyze the next token prediction of RAG given prefix $x_{1:i-1}$ as sampling the shared *Retrieved Concept* $z^*$ from the input retrieved texts list $R = \{r_1, r_2, ..., r_n\}$ ($r_i$ is a retrieved passage), and then predicting $p(x_i|R, x_{1:i-1})$, which can be formalized as:

$$p(x_i|R, x_{1:i-1}) = \int_{\mathcal{Z}} p(x_i|R, x_{1:i-1}, z)p(z|R, x_{1:i-1})\,dz \tag{2}$$

$$= p(x_i|R, x_{1:i-1}, z^*)p(z^*|R, x_{1:i-1}) + \int_{\mathcal{Z}-\{z^*\}} p(x_i|R, x_{1:i-1}, z)p(z|R, x_{1:i-1})\,dz.$$

Equation 2 describes RAG as distribution fusion. The first term is the prediction that is only conditioned on $z^*$, which is the distribution from retrieved texts. The second term is the prediction that marginalizes out all latent concepts except $z^*$, which is the distribution in LLMs.

## 2.3 Formalize and Explain benefit and detriment by Distribution Difference

In this section, we make derivation on the distribution fusion in Equation 2 to formalize the core of this fusion as the subtraction between two terms measured by distribution difference: one is distribution completion, and the other is distribution contradiction. We reveal that in the next token prediction of RAG, distribution completion is the benefit and distribution contradiction is the detriment. Specifically, inspired by (Xie et al., 2021), Equation 2 is transformed as (see Appendix A):

$$p(x_i|R, x_{1:i-1}) = \int_{\mathcal{Z}} p(x_i|R, x_{1:i-1}, z)p(z|R, x_{1:i-1})\,dz \tag{3}$$

$$\propto \int_{\mathcal{Z}} p(x_i|R, x_{1:i-1}, z)p(R, x_{1:i-1}|z)p(z)\,dz \tag{4}$$

$$\propto \int_{\mathcal{Z}} p(x_i|R, x_{1:i-1}, z)\exp(v(z))p(z)\,dz, \quad v(z) = \log\frac{p(R, x_{1:i-1}|z)}{p(R, x_{1:i-1}|z^*)} \tag{5}$$

Define $r_i$ is a passage in the retrieved texts list $R$, we can get (see detailed proof in Appendix B):

$$v(z) = \log\frac{p(R, x_{1:i-1}|z)}{p(R, x_{1:i-1}|z^*)} \approx \log\frac{\prod_{i=1}^{n} O(1)p(r_i|z)}{\prod_{i=1}^{n} O(1)p(r_i|z^*)} \tag{6}$$

$$\to n * \frac{1}{n}\sum_{i=1}^{n}\log\frac{p(r_i|z)}{p(r_i|z*)} = n * \mathbb{E}_{r \sim P_R}\left[\log\frac{p(r|z)}{p(r|z^*)}\right] \tag{7}$$

$$\propto p_R(r)\log\frac{p(r|z)}{p(r|z^*)} = p_R(r)\log\frac{p_R(r)}{p(r|z^*)} - p_R(r)\log\frac{p_R(r)}{p(r|z)} \tag{8}$$

$$= -(\underbrace{\mathrm{KL}(p_R(r)\|p(r|z))}_{\textbf{Distribution Completion: Benefit}} - \underbrace{\mathrm{KL}(p_R(r)\|p(r|z^*))}_{\textbf{Distribution Contradiction: Detriment}}), \tag{9}$$

$p_R(\cdot)$ is the distribution of the retrieved texts, $p(\cdot)$ is the distribution of the LLM's knowledge. $v(z)$ is an important term in distribution fusion because it reflects the proportion between the latent concept from the space of LLMs and from the retrieved texts. Details are in Appendix C.

**Discuss the Benefit and Detriment Based on the Theoretical Results.** In Equation 9, the first term represents the distribution difference (KL divergence) between retrieved texts ($p_R(r)$) and LLM's

knowledge ($p(r|z)$) given the concept $z$, which is sampled from $\mathcal{Z}$ (latent variables in LLM). This can be defined as **distribution completion** that measures how much out-of-distribution knowledge that retrieved texts provide to LLM in the prediction of the token $x_i$. This term is actually the **benefit** for the prediction of token $x_i$ in RAG. This is because that if the retrieved text $r$ is perfect, $p_R(r)$ would be infinitely close to the ground-truth distribution. The larger the difference between $p_R(r)$ and $p(r|z)$, the more the knowledge distribution about $x_i$ in the LLM's knowledge deviates from the ground truth. This means the retrieved texts can provide more valuable out-of-distribution knowledge to the LLM to predict $x_i$, resulting in greater benefit.

Considering that the retrieved texts are not always perfect and may contain incorrect information and noise that contradict the correct knowledge of LLM, the second term corresponds to the **distribution contradiction** can be used to measure this risk. It represents the distribution difference (KL divergence) between the retrieved texts ($p_R(r)$) and LLM's knowledge given the concept $z^*$, which is sampled from retrieved texts ($p(r|z^*)$). $p(r|z^*)$ is the prediction made by LLM conditioned on the concept $z^*$ sampled from the retrieved texts. If the external knowledge in the retrieved texts contradicts LLM's knowledge, $p(r|z^*)$ will deviate from $p_R(r)$ (the actual distribution of the retrieved texts). Therefore, the difference between $p(r|z^*)$ and $p_R(r)$ primarily stems from the LLM's resistance to any external knowledge in the retrieved texts that conflicts with LLM's knowledge. The larger difference indicates the stronger resistance from LLM, and the more confident the LLM is in its pre-trained knowledge, which means the greater the potential detriment caused by the retrieved texts. So this term is actually the **detriment** for the prediction of token $x_i$ in RAG.

**Corollary 1.** *Two terms about distribution difference in Equation 9 measure the benefit and detriment respectively. The subtraction between benefit and detriment describes the trade-off relationship between the value of external knowledge and its potential risk of misleading LLM in the next token prediction without accessing the utility of retrieved texts.*

## 2.4 Actual Effect of RAG can be Predicted at Token Level

Based on above analysis, we successfully formalize the benefit and detriment in next token prediction of RAG by measuring distribution difference. Next, we further explore the method to compare the values of benefit and detriment. Specifically, we derive Theorem 1 and Theorem 2 from Equation 2:

**Theorem 1.** *Define $\mathcal{D} = \|p(x_i|R, x_{1:i-1}) - p_R(x_i|x_{1:i-1})\|_1$ to measure the difference between the distribution of RAG ($p(x_i|R, x_{1:i-1})$) and the distribution of retrieved texts ($p_R(x_i|x_{1:i-1})$) in token prediction of $x_i$ conditioned on prefix $x_{1:i-1}$. Both benefit and detriment are important terms of the upper and lower bounds of $\mathcal{D}$, which can be described as:*

$$\|\Phi\|_1 - \sqrt{2\mathrm{KL}(p_R(r)\|p(r|z^*))} \leq \mathcal{D} \leq \|\Phi\|_1 + \sqrt{2\mathrm{KL}(p_R(r)\|p(r|z^*))}, \tag{10}$$

$$\Phi \approx \alpha \int_{\mathcal{Z}-\{z^*\}} p(x_i|R, x_{1:i-1}, z)\exp\left[-(\underbrace{\mathrm{KL}(p_R(r)\|p(r|z))}_{\text{benefit}} - \underbrace{\mathrm{KL}(p_R(r)\|p(r|z^*))}_{\text{detriment}})\right] p(z)\,dz,$$

in which $\alpha$ is a constant. Our detailed proof of Theorem 1 can be found in Appendix D.

**Theorem 2.** *$\mathcal{D}$ is the difference, so $\frac{1}{\mathcal{D}}$ can be treated as similarity between $p(x_i|R, x_{1:i-1})$ and $p_R(x_i|x_{1:i-1})$. The result of benefit minus detriment is approximately positively correlated with $\frac{1}{\mathcal{D}}$:*

$$\underbrace{\mathrm{KL}(p_R(r)\|p(r|z))}_{\text{benefit}} - \underbrace{\mathrm{KL}(p_R(r)\|p(r|z^*))}_{\text{detriment}} \propto \frac{1}{\mathcal{D}}. \tag{11}$$

Our proof of Theorem 2 and the maximum error analysis of this approximation is in Appendix E.

**Corollary 2.** *The difference between values of benefit and detriment in Equation 11 indicates the extent to which the benefit (value of external knowledge) outweighs the detriment (potential risk of misleading LLM) in the prediction of token $x_i$. This difference is approximately positively correlated with the representation similarity, which is the value that can be predicted.*

Recapping our motivation that aims to build a theory to predict whether the positive impact of the retrieved texts $R$ on $x_i$ (**benefit**) outweighs the potential risk of misleading the LLM (**detriment**). The key for this is comparing the values of benefit and detriment. As stated in Theorem 2, the result of benefit minus detriment is approximately positively correlated with $\frac{1}{\mathcal{D}}$. Therefore, the value of $\frac{1}{\mathcal{D}}$ at which the benefit minus detriment equals zero is an important threshold. A $\frac{1}{\mathcal{D}}$ value greater than

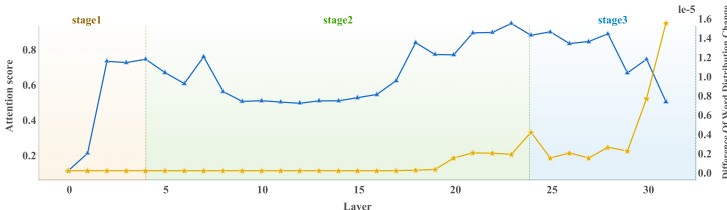

Figure 3: Attention score for $x_i$ (blue line) and difference of word distribution change (yellow line) vary with layers. stage 1: Lexical and Syntactic. stage 2: Text Matching. stage 3: Distribution Fusion.

this threshold indicates that benefit exceeds detriment, while a value less than this threshold indicates that detriment outweighs benefit. We derive Theorem 3 to identify this dividing point and map the value order between benefit and detriment of token $x_i$ to the relationship between representation similarity, which can be calculated in practical applications:

**Theorem 3.** *Define $\mathcal{M} = \|p(x_i|R, x_{1:i-1}) - p(x_i|x_{1:i-1})\|_1$ to measure the difference between distribution of RAG ($p(x_i|R, x_{1:i-1})$) and pure LLM ($p(x_i|x_{1:i-1})$), so $\frac{1}{\mathcal{M}}$ can be treated as the similarity between them. $\frac{1}{\mathcal{D}} = \frac{1}{\mathcal{M}}$ is the dividing point in which benefit is equal to detriment, and the value order between $\frac{1}{\mathcal{D}}$ and $\frac{1}{\mathcal{M}}$ can indicate the value order between benefit and detriment as:*

$$J = \begin{cases} \mathrm{KL}(p_R(r)\|p(r|z)) < \mathrm{KL}(p_R(r)\|p(r|z^*)), \text{ detriment outweighs benefit.} & \text{if } \frac{1}{\mathcal{D}} < \frac{1}{\mathcal{M}} \\ \mathrm{KL}(p_R(r)\|p(r|z)) = \mathrm{KL}(p_R(r)\|p(r|z^*)), \text{ detriment is equal to benefit.} & \text{if } \frac{1}{\mathcal{D}} = \frac{1}{\mathcal{M}} \\ \mathrm{KL}(p_R(r)\|p(r|z)) > \mathrm{KL}(p_R(r)\|p(r|z^*)), \text{ benefit outweighs detriment.} & \text{if } \frac{1}{\mathcal{D}} > \frac{1}{\mathcal{M}} \end{cases} \quad (12)$$

Our detailed proof of Theorem 3 can be found in Appendix F. Equation 12 is a novel principle that can compare the values of benefit and detriment in RAG at token level. It does not rely on additional modules to access the utility of retrieved texts or training but simply compares $\frac{1}{\mathcal{D}}$ and $\frac{1}{\mathcal{M}}$.

**Corollary 3.** *The actual effect of RAG, i.e., the comparison between benefit and detriment, can be predicted at token level by the similarity relationships between $p(x_i|R, x_{1:i-1})$, $p(x_i|x_{1:i-1})$, and $p_R(x_i|x_{1:i-1})$, without additional modules to access the utility of retrieved texts.*

The next section introduces how to apply above theoretical findings to improve RAG in practice.

## 3 TOK-RAG: IMPROVE RAG BASED ON TOKEN-LEVEL THEORY

Tok-RAG is a novel method that enables the LLM and RAG to collaborate at token level for generation to preserve benefit and avoid detriment based on our token-level theory. Tok-RAG makes pure LLM and RAG generate in parallel at token level as shown in Figure 1. It determines which token will be selected by comparing the values of benefit and detriment brought by RAG to the token according to the size relationship between $\frac{1}{\mathcal{D}}$ and $\frac{1}{\mathcal{M}}$ in Equation 12. The terms related to the comparison between $\frac{1}{\mathcal{D}}$ and $\frac{1}{\mathcal{M}}$ consist of three parts: (1) $p(x_i|R, x_{1:i-1})$ can be directly obtained from the prediction of RAG; (2) $p(x_i|x_{1:i-1})$ can be directly obtained from the prediction of pure LLM; (3) however, the distribution of retrieved texts conditioned on the prefix $x_{1:i-1}$, $p_R(x_i|x_{1:i-1})$, is hard to directly obtained, which is the main challenge that the following Section 3.1 aims to solve.

### 3.1 DISTRIBUTION PREDICTION FOR RETRIEVED TEXTS

Our theoretical analysis in Section 2.2 shows that RAG can be modeled as fusing the distribution from retrieved texts with LLMs' distribution. Therefore, an intuitive idea is that the retrieved distribution $p_R(x_i|x_{1:i-1})$ can be approximately predicted by capturing the signal from the retrieved texts in distribution fusion. The main challenges in achieving this are: (1) determining where distribution fusion occurs, and (2) capturing the signal fused from retrieved texts and transforming it to distribution $p_R(x_i|x_{1:i-1})$. To address these challenges, in the following parts, we first explore the operating mechanism of RAG and then propose a novel method that dynamically determines the layers where distribution fusion occurs and use the signal from retrieved texts in these layers as $p_R(x_i|x_{1:i-1})$.

**Exploring the Mechanism of RAG.** This part finds that the mechanism of RAG can be decomposed into two parts. The first is text matching, which means extracting information relevant to the generation of $x_i$ from the retrieved texts $R$. The second is distribution fusion, which means fusing

the distribution from the retrieved texts with the distribution in LLM's knowledge. When performing RAG, LLMs first perform text matching in the middle layers, extracting relevant knowledge from the retrieved texts. As the depth increases, the matching becomes increasingly accurate, reaching a turning point. In the deep layers after this turning point, LLMs instead carry out distribution fusion, and the attention shifts from $R$ to $x_{1:i-1}$. Distribution of $R$ used for fusion comes from the matching information around the turning point (because matching decreases after the turning point). Recapping the two challenges introduced at the beginning of Section 3.1, for the first challenge, we identify the layer where distribution fusion starts by detecting the turning point in Figure 3. For the second challenge, we use the matching information in the layer where distribution fusion starts to approximate the distribution $p_R(x_i|x_{1:i-1})$. Our experiments about these findings are introduced in **Exp 1.** and **Exp 2.**. Experiments are conducted based on LLaMA-2-7B on Natural Question dataset. We also perform these experiments using more LLMs on more datasets. The conclusions of them are consistent with LLaMA-2-7B on Natural Question, and detailed results can be found in Appendix H.

**Exp 1.** For text matching, we quantify the relevance of the information in the retrieved texts to the generation of token $x_i$ by the attention score between token $x_i$ and the tokens in the retrieved texts $R$. We analyze how the sum of attention scores from token $x_i$ to tokens in $R$ varies across layer. As shown by the blue line in Figure 3: (1) The value increases sharply to a peak in shallow layers (0-5), which is mainly because LLMs capture the low-level lexical and syntactic information on the entire input (Tenney et al., 2019). (2) The value first decreases, then increases to a maximum point in the middle layers (5-23), which is mainly because LLMs select the relevant semantics that can be used to generate $x_i$ from $R$ and complete this selection at the maximum point. (3) The value decreases after the maximum point in deep layers (24-32). This is because that LLMs use the selected knowledge at the maximum point for distribution fusion to predict $x_i$, the attention shifts from $R$ to prefix $x_{1:i-1}$.

**Exp 2.** For distribution fusion, since distribution fusion is often marked by a change in word distribution (Bengio et al., 2000), we identify the occurrence of distribution fusion in RAG by comparing the change in word distribution between pure LLM and RAG. (Chuang et al., 2023; Schuster et al., 2022) prove the language heads can be directly applied to the hidden states, so we propose to obtain the word distribution of hidden states in each layer by language heads $\phi$ as $\phi(h_i^l)$, in which $h_i^l$ is the hidden states for token $x_i$ in the $l$-th layer. We then measure the word distribution change in the $l$-th layer by Jensen-Shannon Divergence (JSD) between $\phi(h_i^{l-1})$ and $\phi(h_i^l)$ as: $C = \mathrm{JSD}(\phi(h_i^{l-1})\|\phi(h_i^l))$. The difference of word distribution change between pure LLM and RAG in the $l$-th layer is described as:

$$D^l = |\mathrm{JSD}(\phi(\tilde{h}_i^{l-1})\|\phi(\tilde{h}_i^l)) - \mathrm{JSD}(\phi(h_i^{l-1})\|\phi(h_i^l))|, \tag{13}$$

in which $\tilde{h}_i^{l-1}$ and $\tilde{h}_i^l$ are from RAG, $h_i^{l-1}$ and $h_i^l$ are from pure LLM. The yellow line in Figure 3 shows $D^l$ is very small in the shallow and middle layers (0-23) and rises sharply in the deep layers (24-32). This suggests that distribution fusion occurs in deep layers.

**Dynamically Identify the Layer Where Distribution Fusion Starts.** For $p(x_i|R, x_{1:i-1})$, the layer where distribution fusion starts can be located by detecting the turning point in Figure 3. Specifically, we use $f(l)$ to denote the attention score for $x_i$ and $g(l)$ to denote the difference of word distribution change in Equation 13 varies with layer $l$. The layer where distribution fusion starts is:

$$l^* = \lfloor \frac{1}{2}(\arg\max_l f(l) + \min\{l : g(l) > a\})\rfloor. \tag{14}$$

The first term is the $l$ that maximizes $f(l)$, which is the third turning point in the blue line of Figure 3. The second term means the minimum $l$ value for which $g(l)$ is greater than $a$ (hyperparameter, can be set to 5e-7 according to our statistics), which is the turning point in the yellow line of Figure 3. We take the average of the two values and round down as the layer $l^*$ where distribution fusion starts.

**Matching as Distribution.** The matching information between $R = [rt_1, rt_2, ..., rt_m]$ ($rt$ is the token in $R$) and token $x_i$ at turning point (the $l^*$-th layer) can be used to approximate the distribution $p_R(x_i|x_{1:i-1})$ of the retrieved texts $R$ conditioned on $x_{1:i-1}$. The matching information consists of two parts, one is the attention score, which can measure the matching between retrieved tokens and current token $x_i$ at the hidden state level. The other is the similarity of word embeddings, which can measure the matching between retrieved tokens and current token $x_i$ at the word distribution level:

$$Att = \mathrm{softmax}\left(\frac{(\tilde{h}_i^{l^*}W_q)(\tilde{h}_{1:m}^{l^*}W_k)^T}{\sqrt{d_k}}\right), WordSim = \mathrm{softmax}\left((x_i^{l'-l^*}A)(rt_{1:m}^{l^*}A)^T\right), \tag{15}$$

$W_q$ and $W_k$ are matrices in attention (Vaswani et al., 2017), $\tilde{h}_i^{l*}$ is the hidden state of token $x_i$ and $\tilde{h}_{1:m}^{l*}$ are hidden states of $R$. $A$ is word embedding matrix, $x_i^{l'-l*}$ is the token with the largest logits increase in word distribution from layer $l^*$ to the final layer $l'$, $rt_{1:m}^{l*}$ are tokens in $R$. Then:

$$p_R(x_i|x_{1:i-1}) = \text{softmax}\left(Att \odot WordSim\right), \odot \text{ is element-wise multiplication.} \quad (16)$$

### 3.2 TOKEN-LEVEL COMPARISON BETWEEN BENEFIT AND DETRIMENT IN PRACTICE

Equation 12 shows that the relationship between $\text{Sim}(p(x_i|R, x_{1:i-1}), p_R(x_i|x_{1:i-1}))$ and $\text{Sim}(p(x_i|R, x_{1:i-1}), p(x_i|x_{1:i-1}))$ indicates the value order between benefit and detriment ($\text{Sim}(\cdot, \cdot)$ is the similarity). We propose to use the token semantics as the representation for $p(x_i|R, x_{1:i-1})$, $p_R(x_i|x_{1:i-1})$ and $p(x_i|x_{1:i-1})$ and use cosine to compute the similarity. It not only follows the principle of Equation 12 but also takes into account the semantic similarity, which is more robust in practical applications. Specifically, we use word embedding matrix of LLMs to calculate the weighted average word embedding for $p(x_i|R, x_{1:i-1})$ as $\mathbf{w}_{RAG} = \frac{1}{\sum p'} \sum_{(p', \mathbf{w}) \in \mathbb{V}} p'\mathbf{w}$, for each token in vocabulary $\mathbb{V}$, $p'$ is its logits from $p(x_i|R, x_{1:i-1})$ and $\mathbf{w}$ is its word embedding. We can also use this to get the weighted average word embedding $\mathbf{w}_{LLM}$ for $p(x_i|x_{1:i-1})$ and $\mathbf{w}_{IR}$ for $p_R(x_i|x_{1:i-1})$. The similarity between them can be calculated via cosine similarity as:

$$\text{Sim}(p(x_i|R, x_{1:i-1}), p_R(x_i|x_{1:i-1})) = \cos(\mathbf{w}_{RAG}, \mathbf{w}_{IR}) \quad (17)$$

$$\text{Sim}(p(x_i|R, x_{1:i-1}), p(x_i|x_{1:i-1})) = \cos(\mathbf{w}_{RAG}, \mathbf{w}_{LLM}) \quad (18)$$

Combining our theoretical analysis of Theorem 1, 2 and 3, we can derive this principle to compare the values of benefit and detriment brought by RAG to the token $x_i$ in practical applications:

$$s = \begin{cases} \text{benefit win} & \text{if } \cos(\mathbf{w}_{RAG}, \mathbf{w}_{IR}) \geq \cos(\mathbf{w}_{RAG}, \mathbf{w}_{LLM}), \\ \text{detriment win} & \text{if } \cos(\mathbf{w}_{RAG}, \mathbf{w}_{IR}) < \cos(\mathbf{w}_{RAG}, \mathbf{w}_{LLM}), \end{cases} \quad (19)$$

Given the prefix $x_{1:i-1}$, RAG generates the token $x_i$ and LLM w/o RAG generates $x_i'$. We compare the value of benefit and detriment by Eqn 19. If benefit wins, $x_i$ is selected, otherwise, $x_i'$ is selected. The selected token will be concatenated with $x_{1:i-1}$ as the new prefix for next step generation.

## 4 EXPERIMENTS

### 4.1 EXPERIMENTAL DETAILS

**Practical RAG Performance.** One experiment is in RAG setting for short-form Q&A given retrieved texts with different qualities, it evaluates the robustness and performance of RAG. The other is RAG for many long-form text generation tasks including dialogue, code generation, slot filling, language modeling and long-form Q&A. Baselines include the methods that use additional modules to filter irrelevant texts (**NLI+RAG** (Yoran et al., 2024)) or as action triggers (**CRAG** (Yan et al., 2024)), fine-tune LLMs for robust RAG (**RetRobust** (Yoran et al., 2024) and **INFO-RAG** (Xu et al., 2024b)) and fine-tune LLMs to dynamically retrieve and critique retrieved texts (**Self-RAG** (Asai et al., 2023)).

**Setup for Benefit-Detriment Comparison Experiment.** Given prefix $x_{1:i-1}$ and retrieved texts $R$, our motivation aims to build a theory to predict whether the positive impact of the retrieved texts $R$ on $x_i$ (**benefit**) outweighs the potential risk of misleading LLM (**detriment**). This is a binary classification task at token-level. To evaluate this, we construct test data and ground-truth as:

(1) For a sentence $x$, we truncate it at the $i$-th token to obtain the prefix $x_{1:i-1}$ and the next token $x_i$.
(2) Input prefix $x_{1:i-1}$ to LLM w/o RAG and LLM w/ RAG to get the predicted token $a$ and $b$.
(3) If $b$ is $x_i$ but $a$ not, it means LLM w/ RAG performs better than LLM w/o RAG, so the benefit is greater than detriment, the ground-truth label is 1.
(4) If $a$ is $x_i$ but $b$ not, it means LLM w/o RAG performs better than LLM w/ RAG, so the benefit is lower than detriment, the ground-truth label is 0.
We use the above method to traverse all sentences in the datasets to obtain prefix and its ground-truth as samples. In evaluation, we input prefix to LLM w/ RAG and use our theoretical findings to judge whether the benefit of the next predicted token is greater than detriment. We use AUC and F1-score as the evaluation metrics for this binary classification task.

**Baselines for Benefit-Detriment Comparison Experiment.** Task in benefit-detriment comparison experiment can also be viewed as predicting the correctness of the generated tokens. Therefore, baselines for this are the methods that detect the LLMs' hallucination. We use these baselines to

Table 1: Accuracy on short-form open-domain Q&A given the retrieved texts containing different ratios (0% to 100%) of hard negative passages (irrelevant but are ranked in top-10 by retrieval model). Our Tok-RAG does not need any training or additional modules while baselines need.

| Methods | Train LLM | Utility Evaluator | TriviaQA Ratio of Hard Negative Passages | | | | | | WebQ Ratio of Hard Negative Passages | | | | | | Squad Ratio of Hard Negative Passages | | | | | |
|---|---|---|---|---|---|---|---|---|---|---|---|---|---|---|---|---|---|---|---|---|
| | | | 100% | 80% | 60% | 40% | 20% | 0% | 100% | 80% | 60% | 40% | 20% | 0% | 100% | 80% | 60% | 40% | 20% | 0% |
| Standard RAG | no ✔ | no ✔ | 43.8 | 67.0 | 71.3 | 76.2 | 78.2 | 81.9 | 23.9 | 35.8 | 40.6 | 43.4 | 48.4 | 53.1 | 8.6 | 31.0 | 43.2 | 53.0 | 58.8 | 67.2 |
| NLI+RAG | no ✔ | need ✗ | 50.8 | 61.2 | 68.2 | 73.0 | 76.4 | 79.1 | 30.7 | 40.3 | 44.5 | 47.5 | 50.9 | 52.8 | 9.9 | 21.1 | 33.7 | 43.4 | 51.7 | 60.5 |
| CRAG | no ✔ | need ✗ | 48.2 | 68.3 | 72.5 | 76.7 | 81.5 | 82.2 | 25.6 | 37.4 | 41.9 | 46.2 | 51.5 | 54.9 | 7.4 | 28.7 | 39.6 | 50.7 | 53.2 | 61.1 |
| RetRobust | need ✗ | no ✔ | 49.2 | 67.3 | 72.9 | 77.5 | 79.4 | 82.3 | 30.0 | 38.9 | 42.5 | 48.2 | 49.8 | 54.3 | 10.5 | 30.8 | 43.3 | 52.5 | 58.4 | 66.0 |
| Self-RAG | need ✗ | no ✔ | 43.0 | 68.7 | 73.5 | 76.4 | 80.8 | 82.2 | 18.3 | 34.8 | 42.2 | 47.2 | 51.3 | 57.0 | 5.5 | 27.8 | 38.9 | 46.4 | 52.5 | 58.3 |
| INFO-RAG | need ✗ | no ✔ | 49.7 | 68.4 | 73.2 | 77.9 | 80.0 | 82.5 | 29.7 | 38.0 | 43.9 | 48.1 | 49.4 | 54.8 | 10.7 | 30.1 | 43.5 | 53.7 | 59.2 | 67.5 |
| Tok-RAG (Ours) | no ✔ | no ✔ | **53.5** | **72.9** | **77.6** | **81.3** | **83.4** | **85.7** | **32.9** | **43.8** | **47.3** | **50.0** | **52.9** | **57.3** | **12.8** | **31.3** | **44.5** | **54.1** | **60.8** | **68.1** |

Table 2: Accuracy on various long-form NLP tasks.

| Method | Train LLM | Utility Evaluator | Dialogue Wow F1-Score | Code Generation Python CodeBLEU | Java CodeBLEU | Slot Fill T-REx Accuracy | Language Model WikiText-103 ROUGE | Long-form QA ELI5 ROUGE |
|---|---|---|---|---|---|---|---|---|
| Standard RAG | no ✔ | no ✔ | 7.85 | 21.44 | 22.99 | 55.60 | 60.77 | 15.18 |
| NLI+RAG | no ✔ | need ✗ | 8.04 | 22.79 | 27.45 | 63.28 | 62.05 | 16.14 |
| CRAG | no ✔ | need ✗ | 8.96 | 24.90 | 30.03 | 64.17 | 62.28 | 17.03 |
| RetRobust | need ✗ | no ✔ | 9.03 | 23.18 | 29.74 | 63.19 | 62.40 | 16.90 |
| Self-RAG | need ✗ | no ✔ | 8.55 | 22.15 | 29.60 | 63.24 | 61.22 | 16.47 |
| INFO-RAG | need ✗ | no ✔ | 9.09 | 26.75 | 32.06 | 65.91 | 62.91 | 17.18 |
| Tok-RAG (Ours) | no ✔ | no ✔ | **9.68** | **27.44** | **32.59** | **68.70** | **64.28** | **17.59** |

compare the benefit and detriment by comparing the hallucination degree at token level between RAG and pure LLM (details in Appendix I.3). Baselines include: (1) **Logprobs-based** (Kuhn et al., 2023), we use the value order between top-1 log-probability of the tokens output by pure LLM and RAG to determine the value order between benefit and detriment. (2) **Uncertainty-based**, we use Length-normalized Entropy (Malinin & Gales, 2020) to measure the uncertainty of the tokens and compare it between pure LLM and RAG. (3) **Consistency-based**, we run LLMs multiple times and calculate consistency scores among multiple answers using Lexical and Semantic Similarity (Lin et al., 2022; Chen et al., 2024). We compare these scores between pure LLM and RAG to indicate the comparison between benefit and detriment.

**Implementation details.** As for retrieval in RAG, we follow (Xu et al., 2023) to use ColBERTv2 (Santhanam et al., 2021)las the retriever, and use Wikipedia consisting of 21,015,324 passages (Karpukhin et al., 2020) as retrieval database. **All baselines and Tok-RAG share the same retrieval setup and input.** We use OPT-6.7B, LLaMA-2-7B, and Mistral-7B-v0.1 as LLMs in the benefit-detriment comparison experiment and use greedy-decoding strategy for generation. Details of **retrieval, Tok-RAG, baselines, datasets for each task and metrics** are in Appendix I.

## 4.2 EXPERIMENTAL RESULTS

**Experiment on Practical RAG.** This experiment is under the practical RAG setting for both short-form and long-form NLP tasks and the LLM is LLaMA-2-7B. We adjust the radio of irrelevant passages in the retrieved passage list from 0% to 100%, which can simulate the degree of noise in the retrieved texts. Table 1 shows that for short-form Q&A in RAG given the retrieved texts with various qualities, our Tok-RAG shows better robustness and performance than baselines. Table 2 shows that our Tok-RAG can also perform better than RAG baselines on various long-form NLP tasks. Tok-RAG does not need any additional modules or training and outperforms the strong baselines that need additional modules to evaluate the utility of retrieved texts or training LLMs. This means our Tok-RAG achieves a better trade-off between benefit and detriment in RAG, avoiding detriment while securing benefit. It is because our theoretical analysis helps us propose a more fundamental method in comparing benefit and detriment at token level, while baselines cannot achieve.

**Experiment on Benefit-Detriment Comparison.** Table 3 shows that our Tok-RAG achieves better performance in comparing the values of benefit and detriment at token level in RAG than baselines across different tasks and LLMs. Baselines compare the benefit and detriment by detecting the degree of hallucination, while our Tok-RAG can directly compare the benefit and detriment based on our

Table 3: Performance on comparing the valurs of benefit and detriment at token level. Significant test with p-value $\leq 0.05$ compared with all baselines are denoted as '$+$'. Setup in this experiment is introduced in **Setup for Benefit-Detriment Comparison Experiment** in Section 4.1.

| LLMs | Methods | # Generation Times | Wikitext | | ASQA | | Bio | | NQ | |
|---|---|---|---|---|---|---|---|---|---|---|
| | | | AUC | F1 | AUC | F1 | AUC | F1 | AUC | F1 |
| OPT-6.7B | Logprobs | 2 | 65.25 | 64.33 | 68.96 | 67.55 | 65.24 | 64.59 | 55.31 | 51.41 |
| | Uncertainty | 2 | 64.12 | 63.50 | 66.14 | 63.96 | 65.78 | 64.60 | 56.03 | 52.15 |
| | Consistency-Lexical | 10 | 64.01 | 62.17 | 69.42 | 67.04 | 65.41 | 65.28 | 55.06 | 51.13 |
| | Consistency-Semantic | 10 | 65.93 | 64.22 | 70.11 | 69.50 | 65.76 | 64.37 | 56.24 | 52.88 |
| | Tok-RAG (Ours) | 2 | **68.64**$^+$ | **66.88**$^+$ | **72.28**$^+$ | **72.05**$^+$ | **66.27**$^+$ | **66.04**$^+$ | **57.92**$^+$ | **52.90**$^+$ |
| Mistral-7B | Logprobs | 2 | 73.52 | 72.90 | 68.05 | 66.86 | 65.22 | 64.39 | 57.04 | 57.23 |
| | Uncertainty | 2 | 73.72 | 72.71 | 67.47 | 65.63 | 65.59 | 65.83 | 57.19 | 57.10 |
| | Consistency-Lexical | 10 | 72.15 | 70.44 | 69.16 | 67.33 | 64.79 | 64.33 | 56.95 | 54.37 |
| | Consistency-Semantic | 10 | 73.98 | 72.26 | 70.05 | 69.54 | 65.68 | 65.12 | 57.43 | 56.12 |
| | Tok-RAG (Ours) | 2 | **75.85**$^+$ | **74.11**$^+$ | **71.51**$^+$ | **71.47**$^+$ | **66.37**$^+$ | **66.04**$^+$ | **58.52**$^+$ | **57.56**$^+$ |
| LLaMA-2-7B | Logprobs | 2 | 73.47 | 72.95 | 68.50 | 68.04 | 62.11 | 60.94 | 67.40 | 69.24 |
| | Uncertainty | 2 | 73.98 | 73.01 | 68.72 | 67.63 | 63.67 | 63.50 | 68.03 | 69.15 |
| | Consistency-Lexical | 10 | 73.51 | 71.62 | 70.09 | 68.45 | 62.49 | 61.98 | 68.17 | 70.09 |
| | Consistency-Semantic | 10 | 74.96 | 74.23 | 71.23 | 69.38 | 63.77 | 62.10 | 69.72 | 71.14 |
| | Tok-RAG (Ours) | 2 | **81.89**$^+$ | **80.42**$^+$ | **76.96**$^+$ | **76.80**$^+$ | **64.08**$^+$ | **64.19**$^+$ | **70.50**$^+$ | **72.45**$^+$ |

theoretical analysis, which is more fundamental so it performs better. The detailed setup for this experiment can be found in Section 4.1.

**Analysis on Computational Costs.** Table 4 shows that our Tok-RAG achieves significant performance (accuracy) improvement with little increase of GPU memory and running time. In addition, the baselines all require training LLM or introducing additional modules, while our method is based on our solid theoretical findings, dose not require training LLM nor additional modules. The parallel generation at the token-level can be done in a batch with 2 batch size. In actual practice, this does not bring significant computational time or GPU memory overhead. Experiments are performed on three Q&A datasets (TrviaQA, WebQ, Squad) with V100 GPU, the LLM is LLaMA-2-7B.

| Method | GPU (GB) ↓ | Time (s) ↓ | Per. (Acc) ↑ |
|---|---|---|---|
| Standard | 17.50 | 1.95 | 51.96 |
| NLI+RAG | 20.25 | 2.03 | 49.76 |
| CRAG | 20.60 | 2.15 | 51.53 |
| RetRobust | 17.50 | 2.00 | 52.98 |
| Self-RAG | 17.50 | 2.98 | 50.26 |
| INFO-RAG | 17.50 | 2.10 | 52.35 |
| Tok-RAG | 23.98 | 2.24 | **56.12** |

Table 4: Comparison about GPU Memory, Time, and Performance.

## 5 RELATED WORK

**Robust RAG.** To make LLMs robust in RAG to avoid the detriment caused from noisy in retrieved texts, some methods use additional modules to filter out irrelevant documents (Yoran et al., 2024; Yan et al., 2024; Deng et al., 2023). Some methods train LLMs to make them robust to noisy in retrieved texts (Xu et al., 2024b; Yoran et al., 2024). Some methods let LLMs dynamically determine whether the query needs RAG (Asai et al., 2023; Xu et al., 2023; Ren et al., 2023; Feng et al., 2023; Mallen et al., 2022; Jiang et al., 2023). All the previous works solve the contradiction between benefit and detriment in RAG from the perspective of application but lacking essential and theoretical analysis, which limits the understanding and cannot find the fundamental method to solve it. Our paper explains the benefit and detriment in RAG by theoretical analysis and proposes a novel method to preserve benefit while avoiding detriment without any additional modules or training.

**Theoretical analysis of ICL.** Our paper is inspired by theoretical analysis of ICL. Some works explain ICL as one-step gradient descent (Von Oswald et al., 2023; Akyürek et al., 2022; Dai et al., 2022). Besides, there are other explanations of ICL such as Bayes inference (Xie et al., 2021), Bayes model averaging (Zhang et al., 2023), leaning topic structure (Li et al., 2023) and kernel regression (Han et al., 2023). They focus on explaining why ICL occurs. Our contribution lies in analyzing the benefit and detriment in RAG and proposing a practical method based on our theory.

## 6 CONCLUSIONS AND DISCUSSION

This paper takes the first step in theoretically giving the essential explanation of benefit and detriment in RAG to make them explainable and comparable at token-level. We provide a theory about describing the trade-off between the value of external knowledge and its potential risk of misleading LLMs in next token prediction of RAG. We prove that the actual effect of RAG can be predicted at token level by representation similarity. Based on our theoretical results, we propose a practical novel method that enables pure LLM and RAG to collaborate at token level, gaining benefit while avoiding detriment. Experiments show the effectiveness of our method and support our theoretical results.

ACKNOWLEDGMENTS

This work was supported by the Strategic Priority Research Program of the CAS under Grants No.XDB0680302, the National Natural Science Foundation of China (NSFC) under Grants No. 62276248, and the Youth Innovation Promotion Association CAS under Grants No. 2023111.

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

## A  PROOF FOR EQUATION 5

*Proof.* The transformation is motivated by (Xie et al., 2021) and we apply it to the analysis of RAG:

$$p(x_i|R, x_{1:i-1}) = \int_{\mathcal{Z}} p(x_i|R, x_{1:i-1}, z)p(z|R, x_{1:i-1}) \, dz \tag{20}$$

$$= \int_{\mathcal{Z}} p(x_i|R, x_{1:i-1}, z)\frac{p(R, x_{1:i-1}|z)p(z)}{p(R, x_{1:i-1})} \, dz \tag{21}$$

$$\propto \int_{\mathcal{Z}} p(x_i|R, x_{1:i-1}, z)p(R, x_{1:i-1}|z)p(z) \, dz, \quad p(R, x_{1:i-1}) \text{ is a constant so we drop it} \tag{22}$$

$$\propto \int_{\mathcal{Z}} p(x_i|R, x_{1:i-1}, z)\frac{p(R, x_{1:i-1}|z)}{p(R, x_{1:i-1}|z^*)}p(z) \, dz, \quad \frac{1}{p(R, x_{1:i-1}|z^*)} \text{ is a constant so we add it} \tag{23}$$

$$= \int_{\mathcal{Z}} p(x_i|R, x_{1:i-1}, z)\exp(v(z))p(z) \, dz, \quad v(z) = \log\frac{p(R, x_{1:i-1}|z)}{p(R, x_{1:i-1}|z^*)} \tag{24}$$

$\square$

## B  PROOF FOR EQUATION 6

*Proof.* For $p(R, x_{1:i-1}|z)$ in $v(z) = \log\frac{p(R,x_{1:i-1}|z)}{p(R,x_{1:i-1}|z^*)}$, we can make further derivation as:

$$p(R, x_{1:i-1}|z) = p(x_{1:i-1}|R, z)p(R|z) \tag{25}$$

According to the definition of latent variable model in the analysis of in-context learning from (Xie et al., 2021) that views the latent variable inference as Hidden Markov Model (HMM) and the latent concept $z$ determines the transition probability matrix in HMM hidden states $h$, we can get the following derivations :

$$p(x_{1:i-1}|R, z)p(R|z) = \sum_h p(x_{1:i-1}|h, z)p(h|R, z)p(R|z), \tag{26}$$

$$v(z) = \log\frac{p(R, x_{1:i-1}|z)}{p(R, x_{1:i-1}|z^*)} \tag{27}$$

$$= \log\frac{\sum_h p(x_{1:i-1}|h, z)p(h|R, z)}{\sum_h p(x_{1:i-1}|h, z^*)p(h|R, z^*)} + \log\frac{p(R|z)}{p(R|z^*)}. \tag{28}$$

According to our Assumption 1 that $p(x|h,z^*) > c_1 > 0$, $p(x_{1:i-1}|h,z^*)$ in the denominator of the first term in $v(z)$ is always greater than $c_1$:

$$p(x_{1:i-1}|h,z^*) > c_1.$$

And the numerator $p(x_{1:i-1}|h,z)$ in the numerator of the first term in $v(z)$ is always lower than 1:

$$p(x_{1:i-1}|h,z) < 1.$$

We replace $p(x_{1:i-1}|h,z)$ in the numerator with 1 and $p(x_{1:i-1}|h,z^*)$ in the denominator with $c_1$, and we get:

$$v(z) \le \log\frac{\sum_h 1 \cdot p(h|R,z)}{\sum_h c_1 \cdot p(h|R,z^*)} + \log\frac{p(R|z)}{p(R|z^*)} \tag{29}$$

$$= \log\frac{\sum_h 1 \cdot p(h|R,z)}{\sum_h c_1 \cdot p(h|R,z^*)} + \log\frac{p(R|z)}{p(R|z^*)} \tag{30}$$

$$= -\log c_1 + \log\frac{p(R|z)}{p(R|z^*)} \tag{31}$$

$$= -\log c_1 + \log\frac{\prod_{i=1}^n p(r_i|r_{1:i-1},z)}{\prod_{i=1}^n p(r_i|r_{1:i-1},z^*)}. \tag{32}$$

According to the chain rule, we can transform $p(R,x_{1:i-1}|z)$ as:

$$p(R,x_{1:i-1}|z) = p(x_{1:i-1}|R,z)p(R|z)$$

Eqn 26- 32 shows:

$$v(z) = \log\frac{p(R,x_{1:i-1}|z)}{p(R,x_{1:i-1}|z^*)} \text{ , // Eqn 5.}$$

$$= \log\frac{p(x_{1:i-1}|R,z)p(R|z)}{p(x_{1:i-1}|R,z^*)p(R|z^*)}$$

$$= \log\frac{p(x_{1:i-1}|R,z)}{p(x_{1:i-1}|R,z^*)} + \log\frac{p(R|z)}{p(R|z^*)}$$

$$\le -\log c_1 + \log\frac{p(R|z)}{p(R|z^*)}.$$

So we can get:

$$\log\frac{p(x_{1:i-1}|R,z)}{p(x_{1:i-1}|R,z^*)} \le -\log c_1$$

$$\frac{p(x_{1:i-1}|R,z)}{p(x_{1:i-1}|R,z^*)} \le \frac{1}{c_1}$$

$$p(x_{1:i-1}|R,z) \le \frac{p(x_{1:i-1}|R,z^*)}{c_1}$$

So $p(x_{1:i-1}|R,z)$ is bound by $O(1)$ and then we can get:

$$p(R,x_{1:i-1}|z) = p(x_{1:i-1}|R,z)p(R|z)$$
$$\approx O(1)p(R|z)$$

Since $r_i$ is the passage in $R$, so we can get according to the chain rule:

$$p(R|z) = \prod_{i=1}^n p(r_i|r_{1:i-1},z)$$

So we can get:

$$p(R,x_{1:i-1}|z) = p(x_{1:i-1}|R,z)p(R|z) \approx \prod_{i=1}^n O(1)p(r_i|r_{1:i-1},z) \tag{33}$$

$$\prod_{i=1}^n O(1)p(r_i|r_{1:i-1},z) = \prod_{i=1}^n \sum_{h_{i-1}^d \in \mathcal{D}} p(r_i|h_{i-1}^d,z)p(h_{i-1}^d|r_{1:i-1},z), \tag{34}$$

$r_i$ is a passage in the retrieved texts list $R$, $h_{i-1}^d$ is the hidden state for the delimiter between $r_{i-1}$ and $r_i$ in $R$. Then, we decompose $p(h_{i-1}^d|r_{1:i-1}, z)$ with chain rule:

$$p(h_{i-1}^d|r_{1:i-1}, z) = \sum_h p(h_{i-1}^d|h, r_{1:i-1}, z)p(h|r_{1:i-1}z).$$

According to the standard assumptions of HMM Rabiner (1989), $h_{i-1}^d$ and $r_{1:i-1}$ are conditionally independent, so we can get:

$$p(h_{i-1}^d|r_{1:i-1}, z) = \sum_h p(h_{i-1}^d|h, z)p(h|r_{1:i-1}z).$$

Assumption 2 assumes that for any delimiter hidden state $h^d$ and any other hidden state $h$, the transition probability from $h$ to $h^d$ has upper and lower bound as: $0 \le c_2 \le p(h^d|h, z) \le c_3$. Then we can get:

$$\sum_h p(h_{i-1}^d|h, z)p(h|r_{1:i-1}z) = \sum_h O(1)p(h|r_{1:i-1}z).$$
$$= O(1) \sum_h p(h|r_{1:i-1}z) = O(1).$$

Therefore, $p(h_{i-1}^d|r_{1:i-1}, z) = O(1)$. Then Equation 34 is approximately equal to $\prod_{i=1}^n O(1)p(r_i|z)$, which means that $p(R, x_{1:i-1}|z) \approx \prod_{i=1}^n O(1)p(r_i|z)$, so we can get that:

$$v(z) = \log \frac{p(R, x_{1:i-1}|z)}{p(R, x_{1:i-1}|z^*)} \approx \log \frac{\prod_{i=1}^n O(1)p(r_i|z)}{\prod_{i=1}^n O(1)p(r_i|z^*)} \tag{35}$$

$$\to O(1) + n * \frac{1}{n} \sum_{i=1}^n \log \frac{p(r_i|z)}{p(r_i|z*)} = O(1) + n * \mathbb{E}_{r \sim P_R}\left[\log \frac{p(r_i|z)}{p(r_i|z^*)}\right] \tag{36}$$

$$\propto p_R(r)\log \frac{p(r|z)}{p(r|z^*)} = p_R(r)\log \frac{p_R(r)}{p(r|z^*)} - p_R(r)\log \frac{p_R(r)}{p(r|z)} \tag{37}$$

$$= -(\underbrace{\mathrm{KL}(p_R(r)\|p(r|z))}_{\textbf{benefit},\text{denote as } \Omega} - \underbrace{\mathrm{KL}(p_R(r)\|p(r|z^*)))}_{\textbf{detriment},\text{denote as } \Upsilon}, \tag{38}$$

$p_R(\cdot)$ is the distribution of the retrieved texts, $p(\cdot)$ is the distribution of the LLMs' pre-trianed knowledge.

$\square$

## C EFFECT OF $v(z)$ IN DISTRIBUTION FUSION

Recapping the Equation 38, we find $v(z)$ actually regulates the proportion between LLMs' pre-trained knowledge and retrieved knowledge in distribution fusion of RAG prediction:

- The more benefit outweigh detriment, $v(z) \to -\infty$ and $\exp(v(z)) \to 0$ for all $z \neq z^*$, this indicates that concepts $z$ sampled from LLMs' space contribute little to $p(x_i|R, x_{1:i-1})$. When $z = z^*$, $\exp(r(z^*)) = 1$, which means that latent variable model concentrates more on $z^*$ sampled from retrieved texts. As $v(z)$ decreases, the proportion of retrieved knowledge in becomes larger and larger in fusion.

- The more detriment outweigh benefit, $v(z) \to +\infty$ and $\exp(v(z)) \to +\infty$ for all $z \neq z^*$ and when $z = z^*$, $\exp(r(z^*)) = 1$. This indicates that concepts $z$ sampled from LLMs' space contribute more and more than $z^*$ sampled from retrieved texts as $v(z)$ increases.

## D  PROOF FOR THEOREM 1

*Proof.* Recapping the Equation 2 that describes the distribution fusion in RAG via latent variable model:

$$p(x_i|R, x_{1:i-1}) = \underbrace{\int_{\mathcal{Z}-\{z^*\}} p(x_i|R, x_{1:i-1}, z)p(z|R, x_{1:i-1})\,dz}_{\text{denote as } \Phi} + \underbrace{p(x_i|R, x_{1:i-1}, z^*)p(z^*|R, x_{1:i-1})}_{\text{denote as } \Lambda}.$$

(39)

Since latent concept $z^*$ determines the hidden states $h$, $\Lambda$ can be transformed as:

$$p(x_i|R, x_{1:i-1}, z^*)p(z^*|R, x_{1:i-1}) = \sum_h p(x_i|x_{1:i-1}, h, z^*)p(h|R, x_{1:i-1}.z^*)p(z^*|R, x_{1:i-1}).$$

(40)

Let $p(z^*|R, x_{1:i-1}) = \beta$:

$$p(x_i|R, x_{1:i-1}) = \Phi + \beta \sum_h p(x_i|x_{1:i-1}, h, z^*)p(h|R, x_{1:i-1}.z^*) \tag{41}$$

$$p_R(x_i|x_{1:i-1}) = \sum_h p(x_i|x_{1:i-1}, h, z^*)p_R(h|x_{1:i-1}) \tag{42}$$

$$p_R(h|x_{1:i-1}) \propto p(x_{1:i-1}|h, z^*)p_R(h) \tag{43}$$

$$p(h|R, x_{1:i-1}, z^*) \propto p(x_{1:i-1}|h, z^*)p(h|R, z^*) \tag{44}$$

let probabilities $p(x_i|x_{1:i-1}, h, z^*)p(x_{1:i-1}|h, z^*)$ in Equation 40 is represented as matrix $W \in \mathbb{R}^{|\mathcal{X}| \times |\mathcal{H}|}$ for all possible $x_i \in \mathcal{X}$ and $h \in \mathcal{H}$, $p(h|R, z^*)$ in Equation 44 is matrix $B$, $p_R(h)$ in Equation 43 is $u \in \mathbb{R}^{|\mathcal{H}|}$. We use 1-norm to calculate the difference between $p(x_i|R, x_{1:i-1})$ and $p_R(x_i|x_{1:i-1})$, which can be formalized as:

$$\|p(x_i|R, x_{1:i-1}) - p_R(x_i|x_{1:i-1})\|_1 = \|\Phi + \beta WB - Wu\|_1. \tag{45}$$

Then, according to the triangle inequality of 1-norm, the difference between $p(x_i|R, x_{1:i-1})$ and $p_R(x_i|x_{1:i-1})$ is bouned by:

$$\|\Phi\|_1 - \|\beta WB - Wu\|_1 \le \|\Phi + \beta WB - Wu\|_1 \le \|\Phi\|_1 + \|\beta WB - Wu\|_1. \tag{46}$$

We consider to further analyze $\|\beta WB - Wu\|_1$ inspired by Xie et al. (2021):

$$\|\beta WB - Wu\|_1 = \sum_{i=1}^{|\mathcal{X}|} |W_i^T(\beta B - u)|_i \tag{47}$$

$$= \sum_{i=1}^{|\mathcal{X}|} |\sum_{j=1}^{|\mathcal{H}|} W_{ij}(\beta B - u)_j| \tag{48}$$

$$\le \sum_{i=1}^{|\mathcal{X}|} \sum_{j=1}^{|\mathcal{H}|} W_{ij}|(\beta B - u)_j| \tag{49}$$

$$= \sum_{j=1}^{|\mathcal{H}|} (\sum_{i=1}^{|\mathcal{X}|} W_{ij})|(\beta B - u)_j| \tag{50}$$

$$= \sum_{j=1}^{|\mathcal{H}|} |(\beta B - u)_j| \tag{51}$$

$$= \|\beta B - u\|_1 \tag{52}$$

Then:

$$\|\beta B - u\|_1 = 2TV(p_R(\cdot), \beta p(\cdot|R, z^*)) \quad \text{TV is Total Variation Distance.} \tag{53}$$

$$\le 2\beta TV(p_R(\cdot), p(\cdot|R, z^*)) \tag{54}$$

$$\le \sqrt{2\text{KL}(p_R(\cdot)\|p(\cdot|R, z^*))} \quad \text{Pinsker's Inequality.} \tag{55}$$

$$\le \sqrt{2\text{KL}(p_R(\cdot)\|p(\cdot|z^*))} \tag{56}$$

$$\approx \sqrt{2\text{KL}(p_R(r)\|p(r|z^*))}, \tag{57}$$

in which $r$ is the passage in $R$, $\text{KL}(p_R(r)\|p(r|z^*))$ is actually the **detriment** in Equation 9. Recapping Equation 46, we can get:

$$\|\Phi + \beta WB - Wu\|_1 \leq \|\Phi\|_1 + \sqrt{2\text{KL}(p_R(r)\|p(r|z^*))} \tag{58}$$

Since $0 \leq \|\beta WB - Wu\|_1 \leq 2\sqrt{2\text{KL}(p_R(\cdot)\|p(\cdot|z^*))}$ and $\|\Phi + \beta WB - Wu\|_1 \geq \|\Phi\|_1 - \|\beta WB - Wu\|_1$, then the lower bound for $\|\Phi + \beta WB - Wu\|_1$ is included in:

$$\left[\|\Phi\|_1 - \sqrt{2\text{KL}(p_R(\cdot)\|p(\cdot|z^*))}, \|\Phi\|_1\right], \tag{59}$$

we take the minimum value as the lower bound. Define $\mathcal{D} = \|p(x_i|R, x_{1:i-1}) - p_R(x_i|x_{1:i-1})\|_1$ is the difference between $p(x_i|R, x_{1:i-1})$ and $p_R(x_i|x_{1:i-1})$, according to Equation 45 and 46, the lower and upper bound for $\mathcal{D}$ is:

$$\|\Phi\|_1 - \sqrt{2\underbrace{\text{KL}(p_R(r)\|p(r|z^*))}_{\textbf{detriment}}} \leq \mathcal{D} \leq \|\Phi\|_1 + \sqrt{2\underbrace{\text{KL}(p_R(r)\|p(r|z^*))}_{\textbf{detriment}}}, \tag{60}$$

For ease of description, we denote benefit $\text{KL}(p_R(r)\|p(r|z))$ as $\Omega$ and denote detriment $\text{KL}(p_R(r)\|p(r|z^*))$ as $\Upsilon$. Recapping Equation 5 and 9:

$$p(x_i|R, x_{1:i-1}) = \underbrace{\int_{\mathcal{Z}-\{z^*\}} p(x_i|R, x_{1:i-1}, z)p(z|R, x_{1:i-1})\,dz}_{\text{denote as } \Phi} + \underbrace{p(x_i|R, x_{1:i-1}, z^*)p(z^*|R, x_{1:i-1})}_{\text{denote as } \Lambda}.$$

$$= \int_{\mathcal{Z}} p(x_i|R, x_{1:i-1}, z)p(z|R, x_{1:i-1})\,dz$$

$$= \int_{\mathcal{Z}} p(x_i|R, x_{1:i-1}, z)\frac{p(R, x_{1:i-1}|z)p(z)}{p(R, x_{1:i-1})}\,dz$$

$$\propto \int_{\mathcal{Z}} p(x_i|R, x_{1:i-1}, z)p(R, x_{1:i-1}|z)p(z)\,dz, \quad p(R, x_{1:i-1}) \text{ is a constant so we drop it}$$

$$= \int_{\mathcal{Z}} p(x_i|R, x_{1:i-1}, z)\frac{p(R, x_{1:i-1}|z)}{p(R, x_{1:i-1}|z^*)}p(z)\,dz, \quad \frac{1}{p(R, x_{1:i-1}|z^*)} \text{ is a constant so we add it}$$

$$= \int_{\mathcal{Z}} p(x_i|R, x_{1:i-1}, z)\exp(v(z))p(z)\,dz, \quad v(z) = \log\frac{p(R, x_{1:i-1}|z)}{p(R, x_{1:i-1}|z^*)}$$

$$v(z) = \log\frac{p(R, x_{1:i-1}|z)}{p(R, x_{1:i-1}|z^*)} \approx -\left[\underbrace{\text{KL}(p_R(r)\|p(r|z))}_{\textbf{benefit},\text{denote as } \Omega} - \underbrace{\text{KL}(p_R(r)\|p(r|z^*))}_{\textbf{detriment},\text{denote as } \Upsilon}\right]$$

$\Phi$ in Equation 60 can be transformed as:

$$\Phi = \int_{\mathcal{Z}-\{z^*\}} p(x_i|R, x_{1:i-1}, z)p(z|R, x_{1:i-1})\,dz \tag{61}$$

$$= \frac{p(R, x_{1:i-1}|z^*)}{p(R, x_{1:i-1})}\int_{\mathcal{Z}-\{z^*\}} p(x_i|R, x_{1:i-1}, z)\exp(v(z))p(z)\,dz \tag{62}$$

$$= \alpha\int_{\mathcal{Z}-\{z^*\}} p(x_i|R, x_{1:i-1}, z)\exp(v(z))p(z)\,dz, \quad (p(R, x_{1:i-1}|z^*) \text{ and } p(R, x_{1:i-1}) \text{ are constants})$$
$$\tag{63}$$

$$\approx \alpha\int_{\mathcal{Z}-\{z^*\}} p(x_i|R, x_{1:i-1}, z)\exp(-(\Omega - \Upsilon))p(z)\,dz \quad \text{(Equation 9).} \tag{64}$$

Now the Theorem 1 has been proven.

$\square$

# E  PROOF FOR THEOREM 2

In this section, we try to prove that the gap between values of benefit and detriment is approximately positively correlated with the similarity ($\frac{1}{\mathcal{D}}$) between $p(x_i|R, x_{1:i-1})$ and $p_R(x_i|x_{1:i-1})$. To achieve this, we can start from Equation 60 to prove that the gap between values of benefit and detriment is negatively correlated with the difference ($\mathcal{D}$) between $p(x_i|R, x_{1:i-1})$ and $p_R(x_i|x_{1:i-1})$, which is actually the reciprocal of similarity ($\frac{1}{\mathcal{D}}$). Specifically, we want to prove that the gap between values of benefit and detriment ($\mathrm{KL}(p_R(r)\|p(r|z)) - \mathrm{KL}(p_R(r)\|p(r|z^*))$) is negatively correlated with both lower and upper bound in Equation 60. For ease of description, we denote benefit $\mathrm{KL}(p_R(r)\|p(r|z))$ as $\Omega$ and denote detriment $\mathrm{KL}(p_R(r)\|p(r|z^*))$ as $\Upsilon$.

*Proof.* Recapping Equation 5 and 9:

$$p(x_i|R, x_{1:i-1}) = \underbrace{\int_{\mathcal{Z}-\{z^*\}} p(x_i|R, x_{1:i-1}, z)p(z|R, x_{1:i-1})\, dz}_{\text{denote as } \Phi} + \underbrace{p(x_i|R, x_{1:i-1}, z^*)p(z^*|R, x_{1:i-1})}_{\text{denote as } \Lambda}.$$

$$= \int_{\mathcal{Z}} p(x_i|R, x_{1:i-1}, z)p(z|R, x_{1:i-1})\, dz$$

$$= \int_{\mathcal{Z}} p(x_i|R, x_{1:i-1}, z)\frac{p(R, x_{1:i-1}|z)p(z)}{p(R, x_{1:i-1})}\, dz$$

$$\propto \int_{\mathcal{Z}} p(x_i|R, x_{1:i-1}, z)p(R, x_{1:i-1}|z)p(z)\, dz, \quad p(R, x_{1:i-1}) \text{ is a constant so we drop it}$$

$$= \int_{\mathcal{Z}} p(x_i|R, x_{1:i-1}, z)\frac{p(R, x_{1:i-1}|z)}{p(R, x_{1:i-1}|z^*)}p(z)\, dz, \quad \frac{1}{p(R, x_{1:i-1}|z^*)} \text{ is a constant so we add it}$$

$$= \int_{\mathcal{Z}} p(x_i|R, x_{1:i-1}, z)\exp(v(z))p(z)\, dz, \quad v(z) = \log\frac{p(R, x_{1:i-1}|z)}{p(R, x_{1:i-1}|z^*)}$$

$$v(z) = \log\frac{p(R, x_{1:i-1}|z)}{p(R, x_{1:i-1}|z^*)} \approx - \left[ \underbrace{\mathrm{KL}(p_R(r)\|p(r|z))}_{\textbf{benefit},\text{denote as } \Omega} - \underbrace{\mathrm{KL}(p_R(r)\|p(r|z^*))}_{\textbf{detriment},\text{denote as } \Upsilon} \right]$$

$\Phi$ in Equation 60 can be transformed as:

$$\Phi = \int_{\mathcal{Z}-\{z^*\}} p(x_i|R, x_{1:i-1}, z)p(z|R, x_{1:i-1})\, dz \tag{65}$$

$$= \frac{p(R, x_{1:i-1}|z^*)}{p(R, x_{1:i-1})} \int_{\mathcal{Z}-\{z^*\}} p(x_i|R, x_{1:i-1}, z)\exp(v(z))p(z)\, dz \tag{66}$$

$$= \alpha \int_{\mathcal{Z}-\{z^*\}} p(x_i|R, x_{1:i-1}, z)\exp(v(z))p(z)\, dz, \quad (p(R, x_{1:i-1}|z^*) \text{ and } p(R, x_{1:i-1}) \text{ are constants}) \tag{67}$$

$$\approx \alpha \int_{\mathcal{Z}-\{z^*\}} p(x_i|R, x_{1:i-1}, z)\exp(-(\Omega - \Upsilon))p(z)\, dz \quad \text{(Equation 9).} \tag{68}$$

Therefore, the lower bound of Equation 60 is:

$$\|\Phi\|_1 - \sqrt{2\Upsilon} \approx \alpha\| \int_{\mathcal{Z}-\{z^*\}} p(x_i|R, x_{1:i-1}, z)\exp(-(\Omega - \Upsilon))p(z)\, dz\|_1 - \sqrt{2\Upsilon} \tag{69}$$

$$\propto \exp(-(\Omega - \Upsilon)) - \sqrt{2\Upsilon} \tag{70}$$

and the upper bound of Equation 60 is:

$$\|\Phi\|_1 + \sqrt{2\Upsilon} \propto \exp(-(\Omega - \Upsilon)) + \sqrt{2\Upsilon} \tag{71}$$

Due to both $\Omega$ and $\Upsilon$ being variables, analyzing the result of subtraction between $\Omega$ and $\Upsilon$ under their simultaneous changes is complex. Therefore, we use the "Separation of variables" to simplify our analysis. Specifically, we first assume that one is constant, and then analyze the changes caused by the variation of another:

- Assume $\Omega$ is constant, as the value of $\Omega - \Upsilon$ increases, $\Upsilon$ decreases and the upper bound $\exp(-(\Omega - \Upsilon)) + \sqrt{2\Upsilon}$ also deceases. In the lower bound $\exp(-(\Omega - \Upsilon)) - \sqrt{2\Upsilon}$, since the first term is an exponential function and the second term is a square root function, a decrease of $\Upsilon$ leads to the decrease in the entire lower bound. Therefore, both lower and upper bounds in Equation 60 decrease as $\Omega - \Upsilon$ increases.

- Assume $\Upsilon$ is constant, as the value of $\Omega - \Upsilon$ increases, $\Omega$ increases and the upper bound $\exp(-(\Omega - \Upsilon)) + \sqrt{2\Upsilon}$ deceases. In the lower bound $\exp(-(\Omega - \Upsilon)) - \sqrt{2\Upsilon}$, since the first term is an exponential function and the second term is a square root function, an increase of $\Omega$ leads to the decrease in the entire lower bound. Therefore, both lower and upper bounds in Equation 60 decrease as $\Omega - \Upsilon$ increases.

On behalf of the analysis above, we can derve that both lower and upper bounds in Equation 60 are negatively correlated with the gap between values of benefit and detriment. Therefore, the difference $\mathcal{D}$ between $p(x_i|R, x_{1:i-1})$ and $p_R(x_i|x_{1:i-1})$ is approximately negatively correlated with the gap between values of benefit and detriment.

Then we try to derive the maximum error for "approximation" in this approximate correlation to make it more standard. According to Eqn 70 and 71, the lower bound of $\mathcal{D}$ is:

$$\|\Phi\|_1 - \sqrt{2\Upsilon} \propto \exp(-(\Omega - \Upsilon)) - \sqrt{2\Upsilon},$$

And the upper bound of $\mathcal{D}$ is:

$$\|\Phi\|_1 + \sqrt{2\Upsilon} \propto \exp(-(\Omega - \Upsilon)) + \sqrt{2\Upsilon},$$

Therefore, when the upper and lower bounds are determined, the maximum range of $\mathcal{D}$ is $2\sqrt{2\Upsilon}$ and the average value of $\mathcal{D}$ approaches $\|\Phi\|_1$. Since Equ 68 shows $\|\Phi\|_1$ is an exponential function of $\Upsilon - \Omega$, so the maximum error rate $e_{max}$ is a function of the detriment $\Upsilon$ brought by RAG, which can be described as:

$$e_{max} = \frac{\sqrt{\Upsilon}}{\exp(\Upsilon)}.$$

According to Eqn 9, the detriment $\Upsilon$ is the KL divergence between two distribution, so the value range of $\Upsilon$ is $(0, +\infty)$. So we can get:

$$\lim_{\Upsilon \to +\infty} \frac{\sqrt{\Upsilon}}{\exp(\Upsilon)} = 0.$$

Therefore, the greater the detriment brought by RAG, the smaller the maximum error rate of this correlation and the more robust of our theory. This shows that our method is effective to avoid the detriment brought by RAG at token-level.

Since $\frac{1}{\mathcal{D}}$ can be treated as the similarity between $p(x_i|R, x_{1:i-1})$ and $p_R(x_i|x_{1:i-1})$ and it is approximately positively correlated with the gap between values of benefit and detriment.:

$$\underbrace{\text{KL}(p_R(r)\|p(r|z))}_{\textbf{benefit}} - \underbrace{\text{KL}(p_R(r)\|p(r|z^*))}_{\textbf{detriment}} \propto \frac{1}{\mathcal{D}}. \tag{72}$$

So we have proved that the gap between values of benefit and detriment is approximately positively correlated with $\frac{1}{\mathcal{D}}$ and the maximum error rate of this approximation decreases as the detriment $\Upsilon$ increases and approaches 0.

$\square$

## F   PROOF FOR THEOREM 3

This section aims to prove:

$$J = \begin{cases} \text{KL}(p_R(r)\|p(r|z)) < \text{KL}(p_R(r)\|p(r|z^*)), \text{detriment outweighs benefit.} & \text{if } \frac{1}{\mathcal{D}} < \frac{1}{\mathcal{M}} \\ \text{KL}(p_R(r)\|p(r|z)) = \text{KL}(p_R(r)\|p(r|z^*)), \text{detriment is equal to benefit.} & \text{if } \frac{1}{\mathcal{D}} = \frac{1}{\mathcal{M}} \\ \text{KL}(p_R(r)\|p(r|z)) > \text{KL}(p_R(r)\|p(r|z^*)), \text{benefit outweighs detriment.} & \text{if } \frac{1}{\mathcal{D}} > \frac{1}{\mathcal{M}} \end{cases} \quad (73)$$

in which $\frac{1}{\mathcal{M}}$ is the similarity between $p(x_i|R, x_{1:i-1})$ and $p(x_i|x_{1:i-1})$ (LLMs' pre-trained knowledge), $\frac{1}{\mathcal{D}}$ is the similarity between $p(x_i|R, x_{1:i-1})$ and $p_R(x_i|x_{1:i-1})$ (distribution of retrieved texts)

*Proof.* When benefit is equal to detriment:

$$\text{KL}(p_R(r)\|p(r|z)) - \text{KL}(p_R(r)\|p(r|z^*)) = 0, \quad (74)$$

which means that:

$$p_R(r)\log\frac{p(r|z)}{p(r|z^*)} = 0, \quad (75)$$

since $p_R(r)$ cannot be 0, then:

$$\log\frac{p(r|z)}{p(r|z*)} = 0, \quad (76)$$

$$\frac{p(r|z)}{p(r|z*)} = 1, \quad (77)$$

$$p(r|z) = p(r|z^*), \quad (78)$$

Recapping Equation 2 that $z^*$ is sampled from retrieved texts and $z$ is sampled from LLMs' pre-trained knowledge, Equation 78 indicates that the knowledge of retrieved texts has been involved in LLLs' pre-trained knowledge, so:

$$p(x_i|x_{1:i-1}) = p_R(x_i|x_{1:i-1}), \quad (79)$$

then:

$$\|p(x_i|R, x_{1:i-1}) - p(x_i\|x_{1:i-1})\|_1 = \|p(x_i|R, x_{1:i-1}) - p_R(x_i\|x_{1:i-1})\|_1, \quad (80)$$

which means that $\frac{1}{\mathcal{D}} = \frac{1}{\mathcal{M}}$ is an important dividing point. When $\frac{1}{\mathcal{D}} = \frac{1}{\mathcal{M}}$, we can get that benefit is equal to detriment and $\text{KL}(p_R(r)\|p(r|z)) - \text{KL}(p_R(r)\|p(r|z^*)) = 0$. Equation 72 indicates that the gap between values of benefit and detriment $(\text{KL}(p_R(r)\|p(r|z)) - \text{KL}(p_R(r)\|p(r|z^*)))$ is approximately positively correlated with $\frac{1}{\mathcal{D}}$. Therefore, when $\frac{1}{\mathcal{D}} > \frac{1}{\mathcal{M}}$ we can get that benefit outweighs detriment $(\text{KL}(p_R(r)\|p(r|z)) - \text{KL}(p_R(r)\|p(r|z^*)) > 0)$. When $\frac{1}{\mathcal{D}} < \frac{1}{\mathcal{M}}$ we can get that detriment outweighs benefit $(\text{KL}(p_R(r)\|p(r|z)) - \text{KL}(p_R(r)\|p(r|z^*)) < 0)$. Now the proof of Theorem 3 has been finished. $\qquad \square$

## G   PROOF FOR RAG IS ACTUALLY UNSUPERVISED IN-CONTEXT LEARNING

This section aims to prove that RAG is actually unsupervised ICL from two perspectives. One is that previous studies find that ICL performs gradient descent as meta-optimizer (Von Oswald et al., 2023; Akyürek et al., 2022; Dai et al., 2022). We prove that in this perspective, the distribution of texts in context drives the learning even without explicit input-output supervision. Therefore, the distribution of unsupervised retrieved texts in RAG, which is actually the distribution of context for query, can also drives the learning. Then we can prove that RAG is actually unsupervised in-context learning. The specific proof is:

*Proof.* From the perspective that ICL performs gradient descent as meta-optimizers, ICL can be formalized as the following:

Gradient descent in optimization of linear layers have a dual form of linear attention (Irie et al., 2022; Aizerman et al., 1964), define a liner layer as:

$$f(x) = W_0 x, \quad (81)$$

in which $W_0$ is the initial weight matrix. Given a sequence of historical input vectors $\mathbf{x_i} \in \mathbb{R}^{d_{in}}$ and corresponding error signals $\mathbf{e_i} \in \mathbb{R}^{d_{out}}$, $i \in [1, N]$ obtained by gradient descent, the update of the weight matrix can be represented as:

$$W' = W_0 + \Delta W = W_0 + \sum_i^N \mathbf{e_i} \otimes \mathbf{x_i}. \tag{82}$$

Recap that the linear attention can be formulated as:

$$\text{LinearAttn}(V, K, \mathbf{q}) = \sum_i \mathbf{v}_i(\mathbf{k}_i^T \mathbf{q}). \tag{83}$$

Then the dual form of updated linear layer with new input $\mathbf{x}_{N+1}$ is:

$$f'(x) = (W_0 + \Delta W)\mathbf{x}_{N+1} \tag{84}$$

$$= (W_0 + \sum_i^N \mathbf{e}_i \otimes \mathbf{x}_i)\mathbf{x}_{N+1} \tag{85}$$

$$= W_0 \mathbf{x}_{N+1} + \sum_i^N (\mathbf{e}_i \otimes \mathbf{x}_i)\mathbf{x}_{N+1} \tag{86}$$

$$= W_0 \mathbf{x}_{N+1} + \sum_i^N \mathbf{e}_i \otimes (\mathbf{x}_i^T \mathbf{x}_{N+1}) \tag{87}$$

$$= W_0 \mathbf{x}_{N+1} + \text{LinearAttn}(E, \mathbf{x}_{1:N}, \mathbf{x}_{N+1}) \tag{88}$$

In in-context learning, the attention of a head is:

$$f_{ICL}(\mathbf{q}) = \text{Attn}(V, K, \mathbf{q}) \tag{89}$$

$$= W_V[B' : B]\text{softmax}\left(\frac{W_K[B' : B]^T \mathbf{q}}{\sqrt{d}}\right), \tag{90}$$

in which $\mathbf{q} = W_Q \mathbf{b}$, $\mathbf{b}$ is the input $t$-th token in query, $W_Q, W_K, W_v$ are projection matrices, $B'$ is demonstrations of the context and $X$ is the prefix for $\mathbf{b}$ of the query. To simplify qualitative analysis, we follow (Dai et al., 2022) to estimate the standard attention as relaxed linear attention, achieved by eliminating the softmax function and the scaling factor:

$$f_{ICL}(\mathbf{q}) \approx W_V[B' : B](W_K[B' : B])^T \mathbf{q} \tag{91}$$

$$= W_V B(W_K B)^T \mathbf{q} + W_V X'(W_K B')^T \mathbf{q} \tag{92}$$

$$= W_V B(W_K B)^T \mathbf{q} + \text{LinearAttn}(W_V B', W_K B', \mathbf{q}) \tag{93}$$

According to Equation 88, the dual form of the Transformer attention is:

$$f_{ICL}(\mathbf{q}) \approx W_V B(W_K X)^T \mathbf{q} + \text{LinearAttn}(W_V B', W_K B', \mathbf{q}) \tag{94}$$

$$= W_V B(W_K B)^T \mathbf{q} + \sum_i W_V \mathbf{b}'_i\left((W_K \mathbf{b}'_i)^T \mathbf{q}\right) \tag{95}$$

$$= W_V B(W_K B)^T \mathbf{q} + \sum_i \left((W_V \mathbf{b}'_i) \otimes (W_K \mathbf{b}'_i)\right) \mathbf{q}. \tag{96}$$

Based on above derivation, we have this finding: comparing Equation 86 with Equation 96, we find that $W_V B(W_K B)^T$ is equal to the initial weight matrix $W_0$, which is zero-shot prediction give query prefix $B$ without demonstrations in the context. Besides, $W_V \mathbf{b}'_i$ is equal to $\mathbf{e}_i$. which is the meta-gradient used to update the weighted matrix. $W_K \mathbf{b}_i$ is equal to the historical input vectors:

$$W_V \mathbf{b}'_i = \mathbf{e}_i \tag{97}$$

$$W_K \mathbf{b}'_i = x_i. \tag{98}$$

In the standard gradient descent with loss $\mathcal{L}$, $e_i = -\eta \frac{\partial \mathcal{L}}{\partial y_i}$ and $\eta$ is the learning rate and $y_i = W_i x_i$ is the output of the linear layer using the weight matrix $W_i$ at step $t$ (Irie et al., 2022). So we can get:

$$\mathbf{e}_i = -\eta \frac{\partial \mathcal{L}}{\partial y_i} = -\eta \frac{\partial \mathcal{L}}{\partial W_i x_i} \tag{99}$$

$$= -\eta \frac{\partial \mathcal{L}}{\partial W_i W_K \mathbf{b}'_i}. \tag{100}$$

Therefore:

$$-\eta \frac{\partial \mathcal{L}}{\partial W_i W_K \mathbf{b}'_i} = W_V \mathbf{b}'_i. \tag{101}$$

So the loss $\mathcal{L}$ ca be represented as:

$$\mathcal{L} = -\frac{1}{\eta} \int W_V \mathbf{b}'_i \, d(W_i W_K \mathbf{b}'_i) \tag{102}$$

Equation 97 and 102 show that the supervision signal, both loss and gradient, are directly related to the semantic representation of the tokens ($b_i$) in the demonstration of context. This suggests that the distribution of the text in context is a direct learning signal for in-context learning, without the need for explicit input-output pairs in the demonstration. From the perspective that ICL performs gradient descent as meta-optimizer, the proof has been finished.

The other pespective is from our theoretical results in Equation 5 and 9 that:

$$p(x_i|R, x_{1:i-1}) = \int_{\mathcal{Z}} p(x_i|R, x_{1:i-1}, z) p(z|R, x_{1:i-1}) \, dz \tag{103}$$

$$= \int_{\mathcal{Z} - \{z^*\}} p(x_i|R, x_{1:i-1}, z) p(z|R, x_{1:i-1}) \, dz + p(x_i|R, x_{1:i-1}, z^*) p(z^*|R, x_{1:i-1}). \tag{104}$$

$$\propto \int_{\mathcal{Z}} p(x_i|R, x_{1:i-1}, z) p(R, x_{1:i-1}|z) p(z) \, dz \tag{105}$$

$$= \int_{\mathcal{Z}} p(x_i|R, x_{1:i-1}, z) \exp(v(z)) p(z) \, dz, \quad v(z) = \log \frac{p(R, x_{1:i-1}|z)}{p(R, x_{1:i-1}|z^*)} \tag{106}$$

$$v(z) = \log \frac{p(R, x_{1:i-1}|z)}{p(R, x_{1:i-1}|z^*)} \approx \log \frac{\prod_{i=1}^{n} O(1) p(r_i|z)}{\prod_{i=1}^{n} O(1) p(r_i|z^*)} \tag{107}$$

$$\to O(1) + n * \frac{1}{n} \sum_{i=1}^{n} \log \frac{p(r_i|z)}{p(r_i|z*)} = O(1) + n * \mathbb{E}_{r \sim P_R} \left[ \log \frac{p(r_i|z)}{p(r_i|z*)} \right] \tag{108}$$

$$\propto p_R(r) \log \frac{p(r|z)}{p(r|z^*)} = p_R(r) \log \frac{p_R(r)}{p(r|z^*)} - p_R(r) \log \frac{p_R(r)}{p(r|z)} \tag{109}$$

$$= -(\mathrm{KL}(p_R(r) \| p(r|z)) - \mathrm{KL}(p_R(r) \| p(r|z^*))), \tag{110}$$

We explain Equation 110 from the perspective of the loss function in gradient descent to explain the learning from retrieved texts of LLMs in RAG. $p_R(r)$ is the distribution of retrieved texts and it can serve as ground truth distribution in loss functions. $p(r|z)$ and $p(r|z^*)$ are distribution estimated by LLMs. Two KL-divergence between ground truth distribution $p_R(r)$ and estimated distribution $p(r|z)$ and $p(r|z^*)$ respectively are loss functions. $r$ is the retrieved passage that invariant in generation process, so what contributes to the change of loss function is sampling more and more accurate *retrieved concept* $z^*$ from the retrieved texts. So $\mathrm{KL}(p_R(r) \| p(r|z^*))$ is the actual loss that can be meta-optimized in RAG, in which $p_R(r)$ is the ground truth distribution and $p(r|z^*)$ is the estimated distribution. As this loss decreases, the value of $v(z)$ when $z$ is not equal to $z^*$ also decreases, which means that in Equation 104, the ratio of the knowledge from LLMs' pre-trained distribution decreases and meanwhile the ratio of knowledge from the retrieved texts increases. Lower loss means that the output of RAG is closer to the distribution of retrieved texts, which is actually that LLMs learning the distribution from retrieved texts in input context. Since $p_R(r)$ is the ground truth in this learning but dose not have any explicit input-output supervision like demonstrations in in-context learning, RAG is the **unsupervised** in-context learning and distribution of retrieved texts ($p_R(r)$) is the unsupervised learning signal.

Based on the above two perspectives, we successfully prove that: *The distribution of retrieved passage r in RAG (i.e., $p_R(r)$) can serve as the unsupervised learning signal for LLMs learning from context, even without explicit input-output supervision. RAG is actually unsupervised in-context Learning that fuses the distribution from retrieved texts with LLMs' pre-trained distribution.* □

# H    EXPLORE THE MECHANISM OF RAG

We also perform experiments in Section 3.1 using more LLMs (OPT-6.7B, Mistral-7B) on more datasets (TriviaQA, WebQ and Squad). The experimental results are shown in Table 5 and 6, and the conclusions of them are consistent with Section 3.1.

| Datasets | Value | 0 | 4 | 8 | 12 | 16 | 20 | 24 | 28 | 32 |
|---|---|---|---|---|---|---|---|---|---|---|
| TriviaQA | Attention Score | 0.12 | 0.79 | 0.65 | 0.51 | 0.59 | 0.72 | 0.86 | 0.69 | 0.43 |
| TriviaQA | Word Distribution Change | 0.10 | 0.11 | 0.10 | 0.10 | 0.13 | 0.22 | 0.38 | 0.56 | 0.82 |
| WebQ | Attention Score | 0.10 | 0.69 | 0.65 | 0.54 | 0.55 | 0.70 | 0.82 | 0.64 | 0.40 |
| WebQ | Word Distribution Change | 0.15 | 0.13 | 0.13 | 0.14 | 0.16 | 0.27 | 0.45 | 0.55 | 0.79 |
| Squad | Attention Score | 0.16 | 0.82 | 0.64 | 0.55 | 0.63 | 0.69 | 0.90 | 0.67 | 0.48 |
| Squad | Word Distribution Change | 0.11 | 0.11 | 0.13 | 0.12 | 0.12 | 0.24 | 0.36 | 0.69 | 0.89 |

Table 5: Attention score for and difference of word distribution change vary with layers based on OPT-6.7B on TriviaQA, WebQ and Squad.

| Datasets | Value | 0 | 4 | 8 | 12 | 16 | 20 | 24 | 28 | 32 |
|---|---|---|---|---|---|---|---|---|---|---|
| TriviaQA | Attention Score | 0.05 | 0.64 | 0.50 | 0.45 | 0.60 | 0.69 | 0.81 | 0.65 | 0.37 |
| TriviaQA | Word Distribution Change | 0.08 | 0.10 | 0.11 | 0.15 | 0.19 | 0.30 | 0.42 | 0.65 | 0.82 |
| WebQ | Attention Score | 0.07 | 0.69 | 0.55 | 0.46 | 0.66 | 0.70 | 0.82 | 0.60 | 0.42 |
| WebQ | Word Distribution Change | 0.12 | 0.14 | 0.14 | 0.17 | 0.29 | 0.38 | 0.50 | 0.67 | 0.85 |
| Squad | Attention Score | 0.10 | 0.75 | 0.60 | 0.48 | 0.65 | 0.72 | 0.86 | 0.62 | 0.40 |
| Squad | Word Distribution Change | 0.13 | 0.13 | 0.15 | 0.19 | 0.35 | 0.40 | 0.57 | 0.70 | 0.84 |

Table 6: Attention score for and difference of word distribution change vary with layers based on Mistral-7B on TriviaQA, WebQ and Squad.

# I    EXPERIMENTAL DETAILS

## I.1    RETRIEVAL

As for the retrieval model and retrieval database, for Slot Filling and Language Modeling, we use ColBERTv2, a late-interaction model with excellent generalization ability as the retriever, and use Wikipedia consisting of 21,015,324 passages as retrieval database. For Code Generation, we SCODE-R as code retriever and use deduplicated source codes in CodeSearchNET as retrieval database. For all the above tasks, we give Top-5 retrieved passages to each example. For ELI5, Dialog, we use the list of contextual passages provided in the datasets as the retrieved list (distractor setting). All baselines and our method share the same retrieved documents.

## I.2    TASKS, DATASETS AND METRICS

**Open-domain Question Answering** Open-domain QA is a typical knowledge-intensive task that can directly evaluate the knowledge of LLMs. We use TriviaQA Joshi et al. (2017), Squad Rajpurkar et al. (2016) and WebQuestions (WebQ) as the datasets. We use cover Exact Match (EM) to determine whether the ground truth exactly appears in the output and the accuracy is used as the evaluation metric, following Schick et al. (2023); Xu et al. (2025)

**Slot Filling** Slot filling requires LLMs to output the object entities for the input subject entity and relation. We use a knowledge-intensive dataset T-REx Elsahar et al. (2018). We use the same evaluation metric as Open-domain QA.

**Long-Form Question Answering** Compared with open-domain QA, LFQA is the QA task whose ground truth answer is a relatively long text. We use ELI5 Fan et al. (2019), a knowledge-intensive dataset for LFQA. We use ROUGE-L as the evaluation metric Petroni et al. (2020).

**Dialogue** Dialogue in our experiment focuses on the factual knowledge. We use Wizard of Wikipedia Dinan et al. (2018) (WoW), a knowledge-powered dialogue dataset whose conversation is grounded with knowledge. We use F1 as the evaluation metric Petroni et al. (2020).

**Language Modeling** We use WikiText-103 Merity (2016), a popular dataset for language modeling. We use ROUGE-L as the evaluation metric.

**Code Generation** Code generation aims to generate the code for the given natural language. We use Java and Python in CodeXGLUE Iyer et al. (2018) for this task. We use CodeBLEU Ren et al. (2020) as the evaluation metric.

## I.3 BASELINES

For benefit-detriment comparison experiment that needs methods to determine the value order between benefit and detriment for each token, it is actually a binary classification task (benefit outweigh detriment or not). The mainstream methods in this area are detecting and comparing the degree of hallucination between tokens generated by LLMs (w/o RAG) and RAG. Below we will describe in detail how we apply these baselines to this task.

**Logprobs.** Logprobs can indicate the confidence for LLMs in generating the tokens(Kuhn et al., 2023). We use the value order between top-1 log-probability of the tokens output by pure LLM and RAG to determine the value order between benefit and detriment for these tokens. If the logprobs of tokens generated by RAG is greater than the logprobs of tokens generated by pure LLM, the benefit outweigh the detriment, otherwise the detriment outweigh the benefit.

**Uncertainty.** We use Length-normalized Entropy (Malinin & Gales, 2020) to measure the uncertainty of the tokens generated by pure LLM and RAG respectively. If the uncertainty of tokens generated by RAG is lower than the uncertainty of tokens generated by pure LLM, the benefit outweigh the detriment, otherwise the detriment outweigh the benefit.

**Consistency-Lexical (Lin et al., 2022).** Consistency-based methods make LLMs perform multiple generations for a question and calculate consistency score among multiple answers. If the consistency score of tokens generated by RAG is greater than the consistency score of tokens generated by pure LLM, the benefit outweigh the detriment, otherwise the detriment outweigh the benefit. Lexical-based consistency means calculating consistency score by lexical-similarity among multiple answers. Since the experiment is at token level, we use the number of tokens that are completely consistent in multiple generations as the consistency score.

**Consistency-Semantic (Chen et al., 2024).** We follow (Chen et al., 2024) to use EigenScore to calculate the semantic similarity among hidden states of tokens in multiple generations and use it as the consistency score.

For open-domain Q&A under practical autoregressive generation setting, baselines for this include the methods that introduce additional modules to filter irrelevant passages (**NLI+RAG** (Yoran et al., 2024)) or as action triggers (**CRAG** (Yan et al., 2024)), train more robust LLMs for RAG (**RetRobust** (Yoran et al., 2024) and **INFO-RAG** (Xu et al., 2024b)) and train LLMs to dynamically retrieve and critique retrieved texts (**Self-RAG** (Asai et al., 2023)).

**NLI+RAG.** This method use a Natural Language Inference model to filter the possible irrelevant documents in retrieved results and provide the remaining documents to LLMs for generation. We follow (Yoran et al., 2024) to use a BART-Large model (Lewis et al., 2019) with 407 million parameters trained on the MNLI dataset (Williams et al., 2017). We consider a query-document pair as entailed if the probability for the entailment label is $\geq 0.5$ and filter the documents with probability for the entailment label $< 0.5$.

**CRAG.** This method uses a retrieval evaluator to assess the correctness of retrieved texts trigger different actions based on the evaluation results. One of the actions is using additional google search API for web search, which is unfair for baselines and our method. So we remove this action and use its knowledge refinement strategy for document filtering (Yan et al., 2024).

**RetRobust.** This method fine-tunes LLMs to properly leverage retrieved passages with a mix of relevant and irrelevant contexts (Yoran et al., 2024).

**INFO-RAG.** This method uses unsupervised method to make LLMs learn to use the retrieved texts robustly. It enables LLMs to judge the correctness of the retrieved texts, extract the correct content and revise the wrong content (Xu et al., 2024b).

**Self-RAG.** This method trains LLMs to dynamically retrieve and critique retrieved texts. Self-RAG first decodes a retrieval token to evaluate the utility of retrieval and control a retrieval component. If retrieval is required, LLM calls an external retrieval module to find top relevant documents, using input query and previous generation. If retrieval is not required, LLM continues generation. If retrieval is needed, LLM first generates critique token evaluating whether retrieved documents are relevant and support generation, and then generates continuation conditioned on the retrieved passages (Asai et al., 2023).

### I.4    IMPLEMENTATION DETAILS

All models are run on a V100 GPU with Pytorch (Paszke et al., 2019) and accelerated by DeepSpeed [2]. As for retrieval for RAG, we follow (Xu et al., 2023; 2024b) to use ColBERTv2 (Santhanam et al., 2021), an excellent generalizable model as the retriever, and use Wikipedia consisting of 21,015,324 passages (Karpukhin et al., 2020) as retrieval database. **All baselines and Tok-RAG share the same retrieval setup and prompt.** We use OPT-6.7B, LLaMA-2-7B and Mistral-7B-v0.1 as LLMs in benefit-detriment comparison experiment and use greedy-decoding strategy for generation.

### I.5    ABLATION STUDY.

Figure 4 shows the effectiveness of our dynamic layer selection strategy in Equation 14 and supports our finding that RAG performs matching in middle layers. Figure 4 shows the AUC when $l^*$ in Equation 14 is set as a fixed value from 0 to 32. Our dynamic layer selection strategy (dashed line) is always better than any fixed layers (solid line). Besides, AUC is higher in middle layers, which supports that RAG performs matching in middle layers and the knowledge in retrieved texts is extracted in the turning point. After the turning point, LLMs instead perform distribution fusion, the matching cannot reflect the distribution of retrieved texts, so AUC decreases.

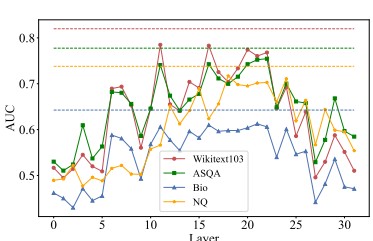

Figure 4: AUC varies with layer.

## J    CASE STUDY

Figure 5 shows the case study for collaborative generation between pure LLM and RAG at token level in our **Tok-RAG**. At the step that pure LLM and RAG generates the different tokens, **Tok-RAG** use our theoretical results in Theorem 3 to compare the benefit and detriment. If benefit is greater than detriment, the token from RAG is selected, otherwise, the token from pure LLM is selected. The selected tokens are marked by green color and bold. Then discarded tokens are marked by gray. The orange arrow represents the direction of token selection and usage. The selected tokens are used for the next step generation of both pure LLM and RAG. This case study visually demonstrates that our **Tok-RAG** effectively enables pure LLM and RAG for collaborative generation to preserve benefit and avoid detriment.

---

[2] https://github.com/microsoft/DeepSpeed

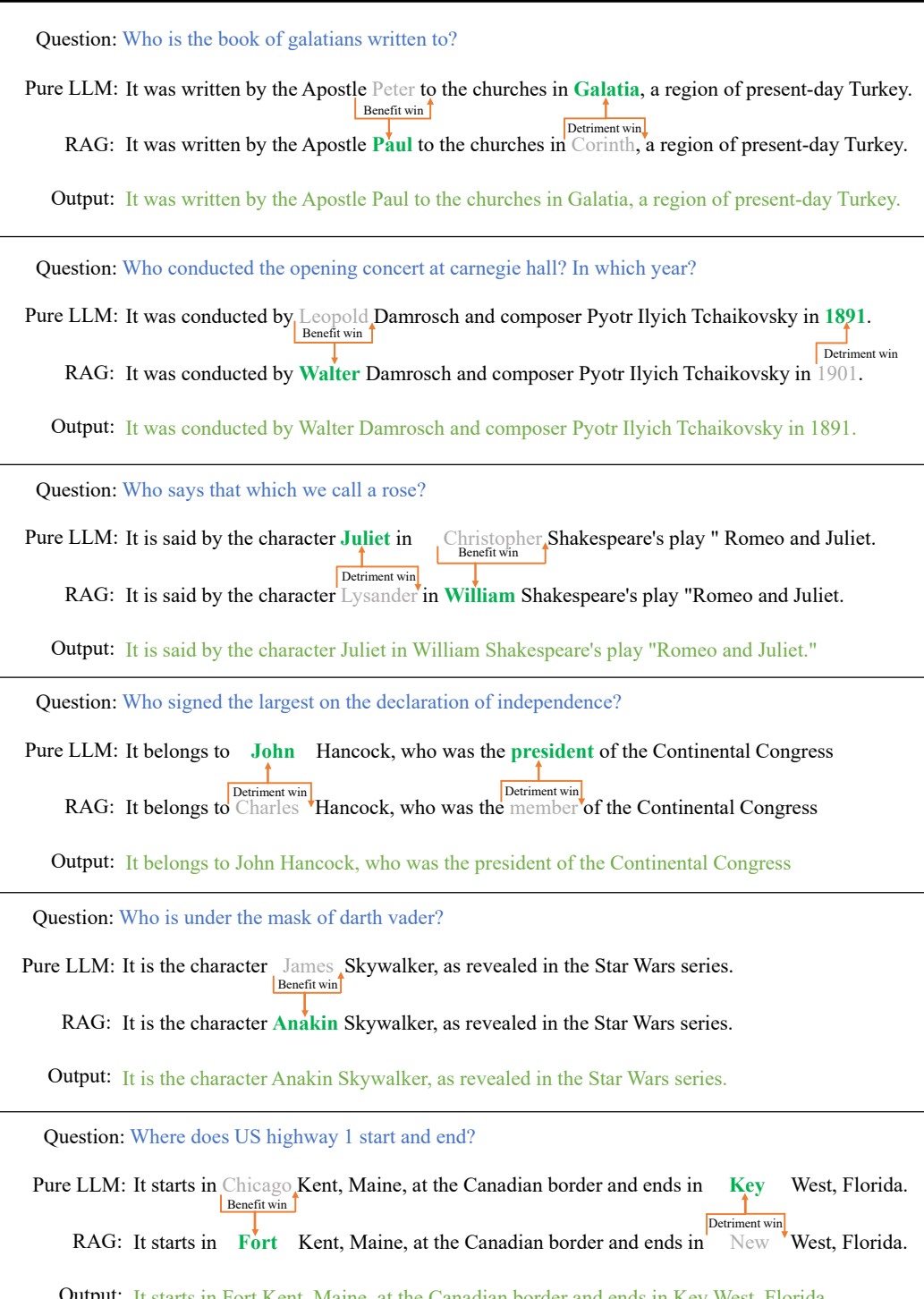

Figure 5: Case study for collaborative generation between pure LLM and RAG at token level in our **Tok-RAG**. Pure LLM and RAG generate the texts in parallel at token level. At the step that pure LLM and RAG generate the different tokens, **Tok-RAG** use our theoretical results in Theorem 3 to compare the benefit and detriment. If benefit is greater than detriment, the token from RAG is selected, otherwise, the token from pure LLM is selected. The selected tokens are marked by green color and bold. The discarded tokens are marked by gray. The orange arrow represents the direction of token selection and usage. The selected tokens are used for the next step generation of both pure LLM and RAG.

