# OpenReview forum: "A Theory for Token-Level Harmonization in Retrieval-Augmented Generation"
_ICLR.cc/2025/Conference — ICLR 2025 Poster_

### Official Review · Reviewer_Rtos · 2024-10-31

**Soundness:** 3
**Presentation:** 2
**Contribution:** 3
**Rating:** 6
**Confidence:** 3

**Summary:**

This paper explores an interesting and important problem of detecting the benefit and detriment of RAG from a theoretical perspective and propose a method called token-RAG. The authors first develop a theory to understand the benefit (retrieval can find valuable external information) and detriment (retrieved documents can be misleading or noisy) at token level. They find that the benefit and detriment can be compared by measuring the relative value of D and M, which is easily calculated in practice. However, calculating p_R for D and M is hard. The authors then introduce a heuristic method to use the attention score and word embedding similarity inside some specific LLM layers to estimate p_R. They finally conduct experiments on several datasets to demonstrate the effectiveness of the proposed token-RAG.

**Strengths:**

- The paper provides a theory to understand the benefit and detriment of the retrieved documents in RAG.
- The paper introduces a method called token-RAG to leverage benefit and prevent detriment.
- The authors conduct experiments on several datasets.

**Weaknesses:**

- The meaning of some notations is not well explained. What is z*?

- Some formulations may be incorrect. Is there any problem with Eq.(2)? Given that p(z|R,x1:{i-1}) is a continuous function, how can you get the term corresponding to p(z|R,x1:{i-1}) out from the integral?

- Why can you go from (4) to (5)? I believe it should not be “=”. Why p(R,x1:{i-1}|z*) is a constant?

- The estimation of p_R(x_i|x_1:{i-1}) seems to be based on heuristics. I am not persuaded that this method can generalize to different datasets and domains.

- In (17) and (18), why use the average word embedding rather than use the real predicted probability of the words? The latter seems to be naturally describing p in M and D.

- The exact token-RAG method is not very clear. From (19), does it mean that if benefit wins, token-RAG will generate the token predicted by RAG, otherwise by LLM directly without RAG?

**Questions:**

See the weakness section.

---

> ### Author Response · Authors · 2024-11-17
> **Author Response (1/2)**
>
> Thank you for your insightful feedback! We follow your suggestions to revise our paper and the new PDF version has been uploaded. The revised content is marked in blue font in our paper. We believe that our response and revisions address your concerns regarding the clarity of our theory and generalizability of our practical method. Following is our response, we look forward to your reply!
>
> ### Response to Weakness
>
> > **W1:** The meaning of some notations is not well explained. What is z*?
>
> **Our Response:** We apologize for the insufficient emphasis on $z^*$ in our paper. $z^*$ is the latent concept that is sampled from retrieved texts. In Line 169-170 of our latest uploaded PDF, we have explained this more clear.
>
> > **W2:** Some formulations may be incorrect. Is there any problem with Eq.(2)? Given that p(z|R,x1:{i-1}) is a continuous function, how can you get the term corresponding to p(z|R,x1:{i-1}) out from the integral?
>
> **Our Response:** Thanks for your question! After our careful verification, Eqn (2) is correct. We can get $p(z|R,x_{1:{i-1}})$ out from the integral because the space of high dimensional concept is **discrete**. The core question for this concern is the about the property of $\mathcal{Z}$, the space of high dimensional concept. In this paper, $\mathcal{Z}$ is a **discrete** set rather than continuous. It is because that the theoretical framework of latent variable inference in our paper is based on Hidden Markov Model (see Line 140-144 of our paper). In Hidden Markov Model, the space of latent concept is a discrete set because it is the standard assumption in HMM [2] and both the transition probability matrix and the observation probability distribution are defined based on a set of discrete states [2].
>
> [2] A Tutorial on Hidden Markov Models and Selected Applications in Speech Recognition
>
> > **W3:** Why can you go from (4) to (5)? I believe it should not be “=”. Why p(R,x1:{i-1}|z*) is a constant?
>
> **Our Response:** We are sorry for this typo, the $=$ is actually $\propto$. In our latest uploaded PDF, we have revised this.

---

> ### Author Response · Authors · 2024-11-17
> **Author Response (2/2)**
>
> > **W4:** The estimation of p_R(x_i|x_1:{i-1}) seems to be based on heuristics. I am not persuaded that this method can generalize to different datasets and domains.
>
> **Our Response:** Thanks for you propose this insightful suggestion to our method more solid. We follow your suggestions to compare our method and baselines in more various tasks (dialogue, code generation, slot filling, language modeling and long-form question answering) in RAG setting and the experimental results shown in the below Table indicate that **our method can also achieve state-of-the-art performance in many other tasks**. In our latest uploaded PDF, we have added this table to the main paper as Table 2 and the specific experimental settings are introduced in Appendix I.
>
>
> |Methods | Training LLM|Utility Evaluator |Dialogue | Code Generation | Code Generation |Slot Filling |Language Modeling |Long-form QA |
> |--------|--------|--------|--------|--------|--------|--------|--------|--------|
> || ||Wow | Python | Java| T-REx | WikiText-103|ELI5|
> || ||F1-Score | CodeBLEU | CodeBLEU| Accuracy | ROUGE|ROUGE|
> |Standard RAG| no | no | 7.85 | 21.44 | 22.99 | 55.60 | 60.77 | 15.18|
> |NLI+RAG| no | need | 8.04 | 22.79 | 27.45 | 63.28| 62.05| 16.14
> |CRAG| no | need |  8.96 | 24.90 | 30.03| 64.17| 62.28 | 17.03
> |RetRobust| need | no |  9.03 | 23.18 | 29.74| 63.19| 62.40 | 16.90|
> |Self-RAG| need | no |  8.55 | 22.15 | 29.60| 63.24| 61.22| 16.47 |
> |INFO-RAG| need | no |   9.09 | 26.75 | 32.06| 65.91 | 62.91 | 17.18 |
> |Tok-RAG (Ours)| no | no | **9.68** | **27.44** | **32.59**| **68.70** | **64.28** | **17.59** |
>
> In this experiment, as for the retrieval model and retrieval database, for Slot Filling and Language Modeling, we use ColBERTv2 , a late-interaction model with excellent generalization ability as the retriever, and use Wikipedia consisting of 21,015,324 passages as retrieval database. For Code Generation, we SCODE-R as code retriever and use deduplicated source codes in CodeSearchNET as retrieval database. For all the above tasks, we give Top-5 retrieved passages to each example. For LFQA, Dialog, we use the list of contextual passages provided in the datasets as the retrieved list (distractor setting). All baselines and our method share the same retrieved documents.
>
> > **W5:** In (17) and (18), why use the average word embedding rather than use the real predicted probability of the words? The latter seems to be naturally describing p in M and D.
>
> **Our Response:** Thanks for your insightful question and please allow us to explain it. Eqn (17) and (18) involves three distributions: distribution of RAG $p(x_i|R,x_{1:i-1})$, distribution of LLM $p(x_i|x_{1:i-1})$ and distribution of retrieved texts $p_R(x_i|x_{1:i-1})$. Although $p(x_i|x_{1:i-1})$ and $p(x_i|R,x_{1:i-1})$ can use the real predicted probability of the words, but $p_R(x_i|x_{1:i-1})$ cannot. So we use the heuristic method in Eqn (16) to calculate $p_R(x_i|x_{1:i-1})$ by attention score and word embedding similarity. Therefore, to ensure that the three distributions are on the same scale, we use average word embedding for $p(x_i|R,x_{1:i-1})$, $p(x_i|x_{1:i-1})$ and $p_R(x_i|x_{1:i-1})$ in Eqn (17) and (18).
>
> > **W6:** The exact token-RAG method is not very clear. From (19), does it mean that if benefit wins, token-RAG will generate the token predicted by RAG, otherwise by LLM directly without RAG?
>
> **Our Response:** Our Tok-RAG method enables the LLM and RAG to collaborate at token level for generation to preserve benefit and avoid detriment. As described in Figure 1 of our paper, given the prefix $x_{1:i-1}$, RAG generates the token $x_i$ and LLM without RAG generates $x_{i}'$. We compare the value of benefit and detriment according to Eqn (19). If benefit wins, $x_i$ is selected, otherwise, $x_{i}'$ is selected. The selected token will be concatenated with $x_{1:i-1}$ as the new prefix for next step generation. This method is derived from our rigorous theoretical analysis in Section 2. In Line 396-398 of our latest uploaded PDF, we explain Equ (19) to make it more clear.

---

> > ### Author Response · Authors · 2024-11-22
> > **Gentle Reminder**
> >
> > Dear Reviewer Rtos
> >
> > Thanks for your response and your insightful suggestions. We are sorry to bother you but since the discussion phase will be ending in a few days, we would like to know can our responses address your concerns and change your assessment of our paper? If you still have any questions or concerns, we are glad to discuss with you!
> >
> > We sincerely appreciate your time and we are looking forward to your reply.
> >
> > Thank you!

---

> > > ### Comment · Reviewer_Rtos · 2024-11-24
> > >
> > > Thank you for your detailed response, which has addressed most of my questions. However, I remain unconvinced by the theoretical justification proposed in this paper. Therefore, I have decided to maintain my original score.

---

> > > > ### Author Response · Authors · 2024-11-25
> > > >
> > > > Dear Reviewer Rtos
> > > >
> > > > Thanks for your feedback! We would like to know what specific concerns you have about our theory? One noteworthy point in our rebuttal is that our method has achieved state-of-the-art results across a diverse range of tasks and datasets, including dialogue, code generation, slot filling, language modeling, and long-form question answering. Importantly, our method is purely grounded in our theoretical framework and does not require any additional training or modules. This further demonstrates the credibility and accuracy of our theory. We hope you will take note of these results!

---

> > > > ### Author Response · Authors · 2024-11-30
> > > > **Hope to get your further reply**
> > > >
> > > > Dear Reviewer Rtos
> > > >
> > > > We are very sorry to bother you. Now the discussion phase will be ending in 2 days, your rating is contrary to other reviewers. We sincerely want to know whether you have any new views of our paper, especially since you mentioned that our response addressed most of your questions? We hope you can notice our new experimental results in rebuttal, which show that method solely based on our theory (without any training or additional filters) can surpass all baselines that require training or additional filters on a variety of tasks and data. This further proves the generalization of our method and the correctness of our theory. We wonder if these can change your score for our paper.
> > > >
> > > > We sincerely appreciate your time and we are looking forward to your reply.
> > > >
> > > > Thank you!

---

> > > > > ### Comment · Reviewer_Rtos · 2024-12-01
> > > > >
> > > > > Thank you. My following-up concern is mainly on this comment "Some formulations may be incorrect. Is there any problem with Eq.(2)? Given that p(z|R,x1:{i-1}) is a continuous function, how can you get the term corresponding to p(z|R,x1:{i-1}) out from the integral?". The author answers that Z is discrete rather than continuous. If so, should we use sum rather than integral in Eq. (2)?

---

> ### Author Response · Authors · 2024-12-01
>
> Thanks for your response. This is a insightful question! In fact, using integral to represent the latent variable inference is a standard formalization that does not vary with the properties of the high-dimensional latent variable space, even if it is discrete [1, 2, 3]. Using sum or integral does not affect the conclusion obtained from our theory.
>
> [1] Language as a Latent Variable: Discrete Generative Models for Sentence Compression
>
> [2] An explanation of in-context learning as implicit bayesian inference
>
> [3] Scalable Inference in Latent Variable Models

---

> > ### Comment · Reviewer_Rtos · 2024-12-01
> >
> > Thank you for your reply. I understand it now. If it is discrete, I would recommend the author use sum in their revision to make it clear.
> >
> > In summary, I think this paper is an interesting paper that proposes a theoretical analysis of RAG and introduces a novel method. Thus, I would like to increase my score.

---

### Official Review · Reviewer_s84W · 2024-11-01

**Soundness:** 3
**Presentation:** 3
**Contribution:** 3
**Rating:** 6
**Confidence:** 4

**Summary:**

This paper provides a theory to explain and trade off the 'benefit' and 'detriment' in next token prediction of RAG. Specifically, the author uses the distribution of the LLM's knowledge and the distribution of retrieved texts to calculate the distribution difference, which can trade off the 'benefit' and 'detriment'. Then, the author employs a series of theoretical analyses and some experiments to explain how to compare the values of 'benefit' and 'detriment'. Finally, the author proves the effectiveness of the method and supports the theoretical findings with experiments.

**Strengths:**

1. This paper connects theory with practice, and the formula derivation is logically clear. Most of the papers are well written and explanatory, and are supported by experiments.
2. The standpoint of this paper is novel, using the perspective of latent variable models to explain RAG, and analyzing the distribution differences between LLM and external knowledge.

**Weaknesses:**

1. In Equation 2, it seems a bit forced to split it into two terms, even though we know that distribution fusion is not a simple addition of distributions.
2. From Equation 4 to 5, the symbol is used incorrectly; An equal cannot be used.
3. In Equation 7, the $P_r$  is not explained. Is it $p_R(r)$?
4. In Section 3.1, during the exploratory experiments on the distribution of retrieved texts $p_R(x_i|x_{1:i-1})$, it is mentioned that this distribution can be approximated using Equations 15 and 16. However, the calculation of the $Att$ score is not clearly explained. It still uses the QKV matrix from LLM? Appendix H only compares different LLMs and datasets, lacking a discussion on the calculation of the $Att$ score. Similarly, the hidden layer state $h$ in Equation 13 also originates from the pre-trained LLM itself.
5. The paper does not evaluate Tok-RAG on LLMs with 33B and 65B scales. Since Tok-RAG generates texts in parallel, it means that two models need to be run at the same time, which may lead to a sharp increase in the demand for computing resources, especially when using larger-scale LLM.

Others:
1. In Equation 2, it is better to write the right side of the equation as $ = p(..) + \int .. dz$ to avoid misunderstandings.
2. There is a mismatch in the notation of log, $ z*$ in Equation 7 compared to Equations 6 and 8. They should be standardized and modified to use $log$ and $z^*$.
3. About table format, the titles of the tables should be placed above the tables. Additionally, the sequence of the tables in the paper goes directly from Table 1 to Table 3. The order of the tables should be revised accordingly.

**Questions:**

1. For the calculation of the $Att$ score, it appears that the LLM is used for inference, and then the $Att$ score is computed for each layer. Essentially, it is still based on the $Wq$ and $W_k$ trained by the LLM itself? This suggests that the distribution of retrieved texts $p_R(x_i|x_{1:i-1})$ has a certain correlation with the knowledge distribution of the LLM $p(x_i|x_{1:i-1})$, and the same applies to word similarity. Is it possible to provide a more rigorous proof of the relationship between  $p_R(x_i|x_{1:i-1})$ and the LLM? For instance, analyzing the relationship between the knowledge distribution $p(x_i,x_{1:i-1})$ of LLM A itself and the retrieval texts distribution $p_R(x_i|x_{1:i-1})$ of LLM B when LLM B performing RAG.
2. In Equation 16, $p_R(x_i|x_{1:i-1})$ is approximated using element-wise multiplication of $Att$ and $WordSim$. Are there other methods that could be used to combine $Att$ and $WordSim$ that could improve the estimation of $p_R(x_i|x_{1:i-1})$. For example, using element-wise addition. You could provide experimental or theoretical explanations.

---

> ### Author Response · Authors · 2024-11-17
> **Author Response (1/2)**
>
> Thank you for your insightful feedback! We follow your suggestions to revise our paper and the new PDF version has been uploaded. The revised content is marked in blue font in our paper. Following is our response, we look forward to your reply!
>
> ### Response to Weakness
>
> > **W1:** In Equation 2, it seems a bit forced to split it into two terms, even though we know that distribution fusion is not a simple addition of distributions.
>
> **Our Response:** Thanks for your feedback. We just use Equation 2 to show the phenomenon of distribution fusion in RAG in a relatively intuitive and easy-to-understand way. The more rigorous and specific description of distributed fusion is given by $v(z)$ in Equation 5. $v(z)$ is an important term in distribution fusion because it reflects the proportion between the latent concept from the space of LLMs and from the retrieved texts. For a detailed explanation, please refer to Appendix C in our paper.
>
>
> > **W2:** From Equation 4 to 5, the symbol is used incorrectly; An equal cannot be used.
>
> **Our Response:** We are sorry for this typo, the $=$ is actually $\propto$. In our latest uploaded PDF, we have revised this.
>
> > **W3:** In Equation 7, the $P_r$ is not explained. Is it $P_R$?
>
> **Our Response:** Yes, $P_r$ should be revised as $P_R$, we are sorry for this typo.
>
> > **W4:** In Section 3.1, during the exploratory experiments on the distribution of retrieved texts $p_R(x_i|x_{1:i-1})$, it is mentioned that this distribution can be approximated using Equations 15 and 16. However, the calculation of the $Att$ score is not clearly explained. It still uses the QKV matrix from LLM? Appendix H only compares different LLMs and datasets, lacking a discussion on the calculation of the $Att$ score. Similarly, the hidden layer state $h$ in Equation 13 also originates from the pre-trained LLM itself.
>
> **Our Response:** Please allow us to explain this. $p_R(x_i|x_{1:i-1})$ means the next token prediction given prefix $x_{1:i-1}$ based on the distribution of retrieved texts $R$. The calculation of the $Att$ score is achieved through the interaction between the prefix $x_{1:i-1}$ and the retrieved text $R$, which measures the contribution of $R$ to the prediction of $x_i$. The parameters, such as QKV matrix, is from LLM but the objects of distribution statistics are the vocabulary and context information in the retrieved text $R$. So our paper calls this method as "approximate" the distribution. Analysis in our response to your question 1 below shows that our method is robust enough to approximate the distribution $p_R(x_i|x_{1:i-1})$, it is less affected by the knowledge distribution of LLM itself.
>
> > **W5:** The paper does not evaluate Tok-RAG on LLMs with 33B and 65B scales. Since Tok-RAG generates texts in parallel, it means that two models need to be run at the same time, which may lead to a sharp increase in the demand for computing resources, especially when using larger-scale LLM.
>
> **Our Response:** Please allow us to clarify it. Although our Tok-RAG needs collaborative generation between pure LLM and RAG, it dose not need two models because it can be achieved by setting batch size as 2 in inference. In a batch, one input is only a query (pure LLM) and the other input is retrieved texts concatenated with the query (RAG). Toke-RAG is achieved in this setting and the GPU memory is not double the number of model parameters (2\*33B or 2\*65B) but only double the number of input and output tensors, which is significantly smaller than model parameters.
>
> > **W6:** In Equation 2, it is better to write the right side of the equation to avoid misunderstandings.
>
> **Our Response:** Thanks for your suggestion, we have made the corresponding revision in our latest uploaded PDF.
>
> > **W7:** There is a mismatch in the notation of log, $z^*$ in Equation 7 compared to Equations 6 and 8. They should be standardized and modified to use $log$ and $z^*$.
>
> **Our Response:** Thanks for your suggestion, we have made the corresponding revision in our latest uploaded PDF.
>
> > **W8:** About table format, the titles of the tables should be placed above the tables. Additionally, the sequence of the tables in the paper goes directly from Table 1 to Table 3. The order of the tables should be revised accordingly.
>
> **Our Response:** Thanks for your suggestion, we have made the corresponding revision in our latest uploaded PDF.

---

> > ### Author Response · Authors · 2024-11-17
> > **Author Response (2/2)**
> >
> > ### Response to Questions
> >
> > > **Q1:** For the calculation of the $Att$ score, it appears that the LLM is used for inference, and then the $Att$ score is computed for each layer. Essentially, it is still based on the $W_q$ and $W_k$ trained by the LLM itself? This suggests that the distribution of retrieved texts $p_R(x_i|x_{1:i-1})$ has a certain correlation with the knowledge distribution of the LLM $p(x_i|x_{1:i-1})$, and the same applies to word similarity. Is it possible to provide a more rigorous proof of the relationship between $p_R(x_i|x_{1:i-1})$ and the LLM? For instance, analyzing the relationship between the knowledge distribution $p(x_i,x_{1:i-1})$ of LLM A itself and the retrieval texts distribution $p_R(x_i|x_{1:i-1})$ of LLM B when LLM B performing RAG.
> >
> > **Our Response:**
> > We follow your suggestions to provide an experiment to show that our method for estimating $p_R(x_i|x_{1:i-1})$ is less influenced by the knowledge distribution of the LLM itself. Specifically, we use 2 different with the same vocabulary (LLaMA-2-7b and Mistral-7b-v0.1). We input the same query $q$ and retrieved texts $R$ to these two different LLMs and use our method to estimate $p_R(x_i|x_{1:i-1})$ from LLaMA and Mistral respectively. We denote $p_R(x_i|x_{1:i-1})$ from LLaMA as $p_{R}^{l}(x_i|x_{1:i-1})$ and $p_{R}(x_i|x_{1:i-1})$ from Mistral as $p_{R}^{m}(x_i|x_{1:i-1})$. We compute the KL divergence between $p_{R}^{l}(x_i|x_{1:i-1})$ and $p_{R}^{m}(x_i|x_{1:i-1})$ over all queries and $R$ of Squad, TriviaQA and WebQA datasets. The experimental results shown in the following talbe indicate that the distribution difference between different LLMs is significantly smaller than difference between different queries. It demonstrates that our method for estimating $p_R(x_i|x_{1:i-1})$ is less influenced by the knowledge distribution of the LLM itself.
> >
> >
> > |Setting | KL | Squad | TriviaQA | WebQ |
> > |--------|--------|--------|--------|--------|
> > |Different LLMs| KL($p_{R}^{l}(x_i$\|$x_{1:i-1})$\|$p_{R}^{m}(x_i$\|$x_{1:i-1})$) | 0.13 | 0.09 |  0.05 |
> > |Different queries and R|  KL($p_{R}^{l}(x_i$\|$x_{1:i-1})$\|$p_{R'}^{l}(x'_i$\|$x'_{1:i-1})$) | 0.75 | 0.68| 0.65 |
> > |Different queries and R|  KL($p_{R}^{m}(x_i$\|$x_{1:i-1})$\|$p_{R'}^{m}(x'_i$\|$x'_{1:i-1})$)| 0.80 | 0.72 | 0.66 |
> >
> >
> > > **Q2:** In Equation 16, $p_R(x_i|x_{1:i-1})$ is approximated using element-wise multiplication of $WordSim$ and $Att$. Are there other methods that could be used to combine $Att$ and $WordSim$ that could improve the estimation of $p_R(x_i|x_{1:i-1})$. For example, using element-wise addition. You could provide experimental or theoretical explanations
> >
> > **Our Response:**
> > We follow your suggestion to conduct ablation study about the combination between $Att$ and $WordSim$:
> >
> > |Combination | Squad | TriviaQA | WebQ |
> > |--------|--------|--------|--------|
> > Element-wise Multiplication| 52.61 | 77.48 | 49.23 |
> > |Element-wise Addition|  50.15 | 76.43 | 48.09|
> > |Element-wise Maximum|  45.28 | 76.19 | 46.70|
> >
> > Element-wise multiplication is a better combination method. Compared with element-wise addition, element-wise multiplication is a nonlinear combination method that emphasizes the intersection probability area between $Att$ and $WordSim$, quickly shrinks the low probability area, and makes the result distribution more concentrated in the high probability area. This can effectively obtain the intersection probability part of $Att$ and $WordSim$ for the distribution $p_R(x_i|x_{1:i-1})$, making our prediction of $p_R(x_i|x_{1:i-1})$ approximate to retrieved texts $R$ itself, and reducing the correlation with the knowledge distribution of different parameters of LLM (word embedding and QKV).

---

> > > ### Author Response · Authors · 2024-11-22
> > > **Gentle Reminder**
> > >
> > > Dear Reviewer s84W
> > >
> > > Thanks for your response and your insightful suggestions. We are sorry to bother you but since the discussion phase will be ending in a few days, we would like to know can our responses address your concerns and change your assessment of our paper? If you still have any questions or concerns, we are glad to discuss with you!
> > >
> > > We sincerely appreciate your time and we are looking forward to your reply.
> > >
> > > Thank you!

---

> ### Comment · Reviewer_s84W · 2024-11-26
>
> Thanks for your response. After careful consideration, I will maintain my current scores.

---

> > ### Author Response · Authors · 2024-11-26
> > **Thanks!**
> >
> > Thank you for your effort in reviewing our paper and for your valuable feedback!

---

### Official Review · Reviewer_aryh · 2024-11-10

**Soundness:** 3
**Presentation:** 2
**Contribution:** 3
**Rating:** 6
**Confidence:** 4

**Summary:**

In RAG settings, retrieved passages may contradict the parametric memory of an LLM (obtained during the LLM training process). The paper explores how to reason and trade-off between the two sources of information in a more optimal manner compared to existing methods. The authors provide a theoretical framework for understanding this in terms of a latent concept variable $z$ and later connect it to a practical method that relies on calculating some of the required quantities by peeking inside the transformer architecture. Without any additional training or models, but with a whitebox approach, the method is able to outperform other RAG baselines that attempt this task.

**Strengths:**

The paper has a fair bit of novel theoretical underpinnings for thinking about retrieved knowledge in terms of a latent concept variable. And if sound and practical, there is significance to the findings especially when it comes to merging two sources of information. The writing is overall clear (except for the items in weaknesses) and quality is fair as well.

**Weaknesses:**

The paper tries to do two things at once, provide both theoretical and practical justification. However, the theoretical justification is at best very convoluted and at worst incorrect. The practical justification is a good start, but the authors test on QA datasets with factoid answers potentially hiding the fact that their method may not generalize outside of this setting. It would be more convincing if the authors did one thing (either theoretical or practical) in a sound and convincing manner rather than attempting both but being unconvincing. In its current form, even if correct and significant, this paper isn't at the acceptance threshold for ICLR. I am willing to change my score if I'm convinced about at least one aspect (theoretical or practical) and that is highlighted in the paper.

1. The theoretical justification needs to be more rigorous and clear (see questions). There are many places where new quantities are defined in an unclear manner which make the proofs hard to follow. There are a couple of assumptions that are likely incorrectly applied or unclearly defined. And terms like "approximate positive correlation" appear in a theorem which to the best of my understanding isn't a standard term. While I suspect the final conclusion that authors draw (Eqn 12) could be practically correct, I am not convinced that the theoretical justification for it is sound.

2. While the authors do not need an external model or a separate training process, they do make a whitebox assumption, i.e. the underlying language model is a transformer and they have access to the layers and logits. This needs to be made the abstract/intro where they downplay other methods but do not mention this limitation of their method.

3. Central to the practical implementation of their method are Eqn 19 and the authors measure performance on factiod QA tasks where the attention distributions are likely to be low entropy (peaky) because one passage likely has the answer and has a strong lexical overlap with the question. However, it isn't clear how much their practical implementation generalizes beyond factiod QA where the attention distribution could be complex and not clearly delineated. For instance, in lines 322-323 the authors claim that "LLMs use the selected knowledge at the maximum point for distribution fusion to predict $x_i$, the attention shifts from $R$ to prefix $x_{1:i-1}$" which seems like pure speculation about the mechanism as we don't have any justification for it and we also don't know that this speculated mechanism generalizes beyond QA tasks.

4. Why do the the authors claim "distribution completion" = "benefit" and "distribution contradiction" = "Detriment"? It could be that the LLM was trained on data that was considered true at the time, but has changed since. In which case the distribution contradiction is beneficial and desirable. The claim about benefit vs detriment is orthogonal to the point of the paper and should be avoid.

5. There is no way a reader can understand this paper without reading the appendix. At least the important assumptions need to be elevated to the main paper, even if the math isn't.

**Questions:**

1. The one question that completely blocked me from understanding the paper was the definition of $p_R$. It first appears in Eqns 36, 37 and is defined as "$p_R(·)$ is the distribution of the retrieved texts". Can the authors more rigorously define this quantity? What is the probability over? i.e. is $p_R(r_i)$ the probability of $r_i$ being retrieved? if $p$ stands for probability of an event, then is it simply $p(r_i)$? Could it just be a uniform distribution then? If not, is there a "retriever" conditioned on a "query" that determines this distribution? In which case, $p_R(r_i) = p(r_i | q)$. And neither of these definitions make sense for Eqn 42, where $p_R(x_i|x_{1:i−1})$ appears. What does it even mean? " distribution of the retrieved texts where the event is x_i conditioned on the previous tokens"? This blocked me from making sense of the proofs beyond Theorem 1.

2. Assumption 1 implies that there exists **some** hidden state $h$ lower bounds it that $p(x|h, z^∗) > c_1 > 0$. Then how is the variable used $c_1$ used in the denominator of Eqn 29 where the sum is over all $h$? If the **some** $h = h^*$, then the denominator can be said to be $c_1 * p(h^*| R, z^*)$ (note the lack of a summation), but this quantity can be arbitrarily small as a function of $z^*$ and $R$. Thus bringing into question the $O(1)$ claim that is made in Eqn 33.

3. Assumption 2 is even more unclear in the definition. Until now there was no mention of the form of $R$ and this is the first time delimiters are introduced. What is the "delimiter hidden state" precisely? Are there any assumptions on input format of $R$. Ideally, in a theoretical justification, the format of the variables should make no difference. It is also very unclear how this assumption leads to $O(1)$ in line 765.

4. It took me a while to understand how Eqn 33 could be potentially derived from Eqn 32. It would be prudent to be more rigorous here for future readers.

5. Even if I assume theorem 1 is correct, in theorem 2, authors claim "approximate positive correlation", but I don't think this is a standard or well defined term. In lines 978 - 983, the authors go from upper and lower bounds to the claim on "approximate correlation" which doesn't seem theoretically sound.

6. Just do double check, is Eqn 29 the same as Eqn 30?

Overall, the paper would be a lot easier to understand and review if there weren't so many leaps of logic in the proofs.

---

> ### Author Response · Authors · 2024-11-17
> **Author Response (1/5)**
>
> Thank you for your insightful feedback! We follow your suggestions to revise our paper and the new PDF version has been uploaded. The revised content is marked in blue font in our paper. We believe that our response and revisions address your concerns regarding the clarity of our theory and generalizability of our practical method.
> - As for the theory, we follow your suggestions to make the definitions and assumptions more clear and add them to the main paper. We answer every one of your questions and solve each of them in the new PDF version to make them easier for readers to understand.
> - As for the practical method, the main concern is about the generalization of our method to more tasks. We follow your suggestions to compare our method with baselines on more tasks such as dialogue, code generation, slot filling, language modeling and long-form question answering. **Experimental results further show that our method can also achieve state-of-the-art performance in many other tasks besides QA.** These experiments are highlighted in Table 2 of the new PDF version. The method based on our theory has shown excellent performance in various tasks without additional training (while baselines need), which further confirms the correctness of our theory.
>
> Following is our response. We look forward to discussing with you!
> ### Response to Weakness
> > **W1:** The theoretical justification needs to be more rigorous and clear (see questions). There are many places where new quantities are defined in an unclear manner which make the proofs hard to follow. There are a couple of assumptions that are likely incorrectly applied or unclearly defined. And terms like "approximate positive correlation" appear in a theorem which to the best of my understanding isn't a standard term. While I suspect the final conclusion that authors draw (Eqn 12) could be practically correct, I am not convinced that the theoretical justification for it is sound.
>
> **Our Response:** We appreciate your efforts to make our theory more understandable and sound, and apologize for the inconvenience our writing has caused you. Regarding this weakness you mentioned, we have responded it in **Our Response to Questions** below. We answer every one of your questions and solve each of them in the new PDF version to make them easier for readers to understand. We sincerely hope that our responses can make our theoretical analysis clearer and change your thoughts to our paper.
>
> > **W2:** While the authors do not need an external model or a separate training process, they do make a whitebox assumption, i.e. the underlying language model is a transformer and they have access to the layers and logits. This needs to be made the abstract/intro where they downplay other methods but do not mention this limitation of their method.
>
> **Our Response:** We appreciate the reviewer for pointing out this point. Please allow us to clarify this. In fact, not only our method, but all the methods in RAG baseline that need to fine-tune LLMs, such as Retrobust, Self-RAG, and INFO-RAG, face the same problem, i.e. needing to access the parameters of LLM and updating them. Compared to these baselines, our method just needs to access LLMs' parameters at inference time without updating them and achieves better performance (Table 1 in our paper). We follow your advice to discuss this point in Line 92-94 of intro.

---

> ### Author Response · Authors · 2024-11-17
> **Author Response (2/5)**
>
> > **W3:** Central to the practical implementation of their method are Eqn 19 and the authors measure performance on factiod QA tasks where the attention distributions are likely to be low entropy (peaky) because one passage likely has the answer and has a strong lexical overlap with the question. However, it isn't clear how much their practical implementation generalizes beyond factiod QA where the attention distribution could be complex and not clearly delineated. For instance, in lines 322-323 the authors claim that "LLMs use the selected knowledge at the maximum point for distribution fusion to predict $x_i$, the attention shifts from $R$ to prefix $x_{1:i-1}$" which seems like pure speculation about the mechanism as we don't have any justification for it and we also don't know that this speculated mechanism generalizes beyond QA tasks.
>
> **Our Response:** Thanks for you propose this insightful suggestion to our method more solid. We follow your suggestions to compare our method and baselines in more various tasks (dialogue, code generation, slot filling, language modeling and long-form question answering) in RAG setting and the experimental results shown in the below Table indicate that **our method can also achieve state-of-the-art performance in many other tasks besides QA**. In our latest uploaded PDF, **we have added this table to the main paper as Table 2 and the specific experimental settings are introduced in Appendix I**.
>
>
> |Methods | Training LLM|Utility Evaluator |Dialogue | Code Generation | Code Generation |Slot Filling |Language Modeling |Long-form QA |
> |--------|--------|--------|--------|--------|--------|--------|--------|--------|
> || ||Wow | Python | Java| T-REx | WikiText-103|ELI5|
> || ||F1-Score | CodeBLEU | CodeBLEU| Accuracy | ROUGE|ROUGE|
> |Standard RAG| no | no | 7.85 | 21.44 | 22.99 | 55.60 | 60.77 | 15.18|
> |NLI+RAG| no | need | 8.04 | 22.79 | 27.45 | 63.28| 62.05| 16.14
> |CRAG| no | need |  8.96 | 24.90 | 30.03| 64.17| 62.28 | 17.03
> |RetRobust| need | no |  9.03 | 23.18 | 29.74| 63.19| 62.40 | 16.90|
> |Self-RAG| need | no |  8.55 | 22.15 | 29.60| 63.24| 61.22| 16.47 |
> |INFO-RAG| need | no |   9.09 | 26.75 | 32.06| 65.91 | 62.91 | 17.18 |
> |Tok-RAG (Ours)| no | no |  **9.68** | **27.44** | **32.59**| **68.70** | **64.28** | **17.59** |
>
> In this experiment, as for the retrieval model and retrieval database, for Slot Filling and Language Modeling, we use ColBERTv2 , a late-interaction model with excellent generalization ability as the retriever, and use Wikipedia consisting of 21,015,324 passages as retrieval database. For Code Generation, we SCODE-R as code retriever and use deduplicated source codes in CodeSearchNET as retrieval database. For all the above tasks, we give Top-5 retrieved passages to each example. For ELI5, Dialog, we use the list of contextual passages provided in the datasets as the retrieved list (distractor setting). All baselines and our method share the same retrieved documents.
>
> > **W4:** Why do the the authors claim "distribution completion" = "benefit" and "distribution contradiction" = "Detriment"? It could be that the LLM was trained on data that was considered true at the time, but has changed since. In which case the distribution contradiction is beneficial and desirable. The claim about benefit vs detriment is orthogonal to the point of the paper and should be avoid.
>
> **Our Response:**
> We invite you to review Line 211-219 in our paper, which is our discussion about the connection between distribution completion (contradiction) and benefit (detriment).
>
> - Distribution completion in our Eqn 9 measures how much **out-of-distribution** knowledge that retrieved texts provide to LLM in the prediction of the token, which can be defined as benefit.
> - Distribution contradiction in our Eqn 9 measures the LLM’s **resistance** to any external knowledge in the retrieved texts that conflicts with LLM's knowledge, which can be defined as detriment.
>
> Our Eqn 9 shows that the next token prediction in RAG is governed by the **combined effect** of benefit and detriment, whose relationship is described as subtraction (distribution completion - distribution contradiction).
>
> As for your concern that "It could be that the LLM was trained on data that was considered true at the time, but has changed since. In which case the distribution contradiction is beneficial and desirable", this situation has already been measured by our distribution completion rather than distribution contradiction. Our distribution contradiction focuses on LLM's **resistance** to external knowledge rather than just difference.

---

> ### Author Response · Authors · 2024-11-17
> **Author Response (3/5)**
>
> > **W5:** There is no way a reader can understand this paper without reading the appendix. At least the important assumptions need to be elevated to the main paper, even if the math isn't.
>
> **Our Response:** We appreciate your suggestions on the readability of the paper. In Line 121-154 of our latest uploaded PDF file, we follow your suggestions to move the important definitions and assumptions to the main body of the paper, ensuring that the core concepts are clearly presented. The mathematical derivations remain in the appendix to maintain the flow of the main text. We believe this change will improve the clarity and comprehensibility of the paper.
>
> ### Response to Questions
>
> > **Q1:** The one question that completely blocked me from understanding the paper was the definition of $p\_{R}$. It first appears in Eqns 36, 37 and is defined as "$p\_{R}()$" is the distribution of the retrieved texts". (**Q1.1**) Can the authors more rigorously define this quantity? (**Q1.2**) What is the probability over? i.e. is $p\_{R}(x\_i)$ the probability of $r_i$ being retrieved? if $p$ stands for probability of an event, then is it simply $p(r\_i)$? (**Q1.3**) Could it just be a uniform distribution then? If not, (**Q1.4**) is there a "retriever" conditioned on a "query" that determines this distribution? In which case, $p\_{R}(r\_i)=p(r\_i|q)$. And neither of these definitions make sense for Eqn 42, where $p\_{R}(x\_i|x\_{i-1})$ appears. (**Q1.5**) What does it even mean?" distribution of the retrieved texts where the event is $x_i$ conditioned on the previous tokens"? This blocked me from making sense of the proofs beyond Theorem 1.
>
> **Our Response:**
> Thanks for your valuable suggestions, please allow us to answer your questions below. We follow your suggestions to make $p\_{R}(\cdot)$ more clear in Line 127-139 of our latest uploaded PDF.  (*We found that OpenReview's rendering of latex formulas is unstable (depending on the browser version), so the display of the following formulas may be abnormal. We recommend you to read Line 127-139 of our latest uploaded paper.*)
>
> > **Q1.1:** More rigorously definition of "$p\_{R}(\cdot)$ is the word distribution of retrieved passages list $R$".
>
> **A1.1:** . $p\_{R}(\cdot)$ is the language modeling distribution just like the distribution $p(\cdot)$ of the LLM. This distribution $p(\cdot)$ is learned through training LLM on large corpus with next-token prediction paradigm. $p(x\_i | x\_{1:i-1})$ represents the probability distribution predicted by the LLM for $x\_i$, given the prefix $x\_{1:i-1}$. The LLM is capable of generating reasonable and coherent text based on the distribution patterns of vocabulary and context learned from the large training data. **Here, $p\_{R}(\cdot)$ refers to the distribution $p(\cdot)$ learned when the training data that is limited to the retrieved passages list $R$**, and it represents the vocabulary and contextual distribution patterns of the retrieved passages list $R$.
>
> > **Q1.2:** What is the probability $p\_{R}(\cdot)$ over?
>
> **A1.2:** $r\_i$ is the $i$-$th$ passage in the retrieved passages list $R$. According to our **A1.1**, $p\_{R}(\cdot)$ is the language modeling distribution, so $p\_{R}(r\_i)$ represents the joint probability of the natural language sequence $r\_{i} = [r\_i^1,r\_i^2,r\_i^3,...,r\_i^l]$ under the distribution $p_{R}({\cdot})$, in which $r^x\_i$ is the $x$-$th$ token in passage $r\_i$. Specifically, according to the chain rule, $p\_{R}(r\_i)$ can be decomposed into the product of a series of conditional probabilities:
> $$
> \begin{align}
> p\_{R}(r\_i)=p\_{R}(r^1\_i) \cdot p\_{R}(r^2\_i|r^1\_i) \cdot p\_{R}(r^3\_i|r^1\_i,r^2\_i) \cdot \cdot \cdot \cdot p\_{R}(r^l\_i|r^1\_i,r^2\_i,r^3\_i,..., r^{l-1}\_i),
> \notag
> \end{align}$$
>
> > **Q1.3:** Could it just be a uniform distribution then?
> > **Q1.4:** is there a "retriever" conditioned on a "query" that determines this distribution?
>
> **A1.3, A1.4:** According to our **A1.1** and **A1.2**, $p_{R}(\cdot)$ is the language modeling distribution and $p\_{R}(r\_i)$ represents the joint probability of the natural language sequence $r\_i$. This distribution is determined by the retrieved texts list $R$, not the retriever.
>
> > **Q1.5:** What does $p\_{R}(x\_i|x\_{1:i-1})$ in Equation 42 mean?
>
> **A1.5:** According to our **A1.1** and **A1.2**, $p\_{R}(\cdot)$ is the language modeling distribution, so $p\_{R}(x\_i|x\_{1:i-1})$ means the next token prediction for $x\_i$ given prefix $x\_{1:i-1}$.

---

> ### Author Response · Authors · 2024-11-17
> **Author Response (4/5)**
>
> > **Q2:** Assumption 1 implies that there exists some hidden state $h$ lower bounds it that $p(x|h,z^*)>c\_{1}>0$. Then how is the variable used $c\_1$ used in the denominator of Eqn 29 where the sum is over all $h$? If the some $h=h^*$, then the denominator can be said to be $c\_{1}\*p(h^*|R,z^*)$, but this quantity can be arbitrarily small as a function of $z^*$ and $R$. Thus bringing into question the $O(1)$ claim that is made in Eqn 33.
>
> **Our Response:**
> There is some misunderstanding here. Please allow us to clarify it, the following part has also been revised in Line 738-783 of our latest uploaded PDF to make this point clear.  (*We found that OpenReview's rendering of latex formulas is unstable (depending on the browser version), so the display of the following formulas may be abnormal. We recommend you to read Line 738-783 of our latest uploaded paper.*)
>
> In Eqn 28, we get:
> $$\begin{align}
>     v(z) = \textrm{log}\frac{\sum\_{h}p(x\_{1:i-1}|h,z)p(h|R,z)}{\sum\_{h}p(x\_{1:i-1}|h,z^*)p(h|R,z^*)} + \textrm{log}\frac{p(R|z)}{p(R|z^*)}.
> \notag\end{align}$$ According to our Assumption 1 that $p(x|h,z^*)>c\_{1}>0$, $p(x\_{1:i-1}|h,z^*)$ in the denominator of the first term in $v(z)$ is always greater than $c\_1$:
> $$\begin{align}
>     p(x\_{1:i-1}|h,z^*) > c\_1.
> \notag\end{align}$$ And the numerator $p(x\_{1:i-1}|h,z)$ in the numerator of the first term in $v(z)$ is always lower than $1$:
> $$\begin{align}
>     p(x\_{1:i-1}|h,z) < 1.\notag
> \end{align}$$ We replace $p(x\_{1:i-1}|h,z)$ in the numerator with $1$ and $p(x\_{1:i-1}|h,z^*)$ in the denominator with $c\_1$, and we get
> $$\begin{align}
>     v(z) &= \textrm{log}\frac{\sum\_{h}p(x\_{1:i-1}|h,z)p(h|R,z)}{\sum\_{h}p(x\_{1:i-1}|h,z^*)p(h|R,z^*)} + \textrm{log}\frac{p(R|z)}{p(R|z^*)} \notag\\\\
>     & \leq  \textrm{log}\frac{\sum\_{h}1 \cdot p(h|R,z)}{\sum\_{h}c_1  \cdot p(h|R,z^*)} + \textrm{log}\frac{p(R|z)}{p(R|z^*)} \text{  , // greater numerator and lower denominator}
> \notag\end{align}$$ In this way, we use $c\_1$ to successfully transform Eqn 28 to get Eqn 29. **This can answer your core question about the usage of $c\_1$**. As for the $O(1)$ claim made in Eqn 33, please see our response to **Q4** below. **In Line 738-783 of our latest uploaded PDF, we have made the corresponding improvements to make our derivation process clearer.**
>
> > **Q3:** Assumption 2 is unclear in the definition. Until now there was no mention of the form of $R$ and this is the first time delimiters are introduced. What is the "delimiter hidden state" precisely? Are there any assumptions on input format of $R$. Ideally, in a theoretical justification, the format of the variables should make no difference. It is also very unclear how this assumption leads to $O(1)$ in line 765.
>
> **Our Response:**
> We apologize for any inconvenience caused by our writing, please allow us to clarify it. The following part has also been revised in Line 122-126, Line 150-154 and Line 795-810 of our latest uploaded PDF to make this point clear.   (*We found that OpenReview's rendering of latex formulas is unstable (depending on the browser version), so the display of the following formulas may be abnormal. We recommend you to read Line 122-126, Line 150-154 and Line 795-810 of our latest uploaded paper.*)
>
> > **Q3.1:** What is the "delimiter hidden state" precisely and what is the input format of $R$?
>
> **A3.1:**
> $R$ is the list that contains many retrieved passages, the format of $R$ is like:
> $$\begin{align}
> R = [r\_1,r\_2,r\_3,...,r\_n],
> \notag\end{align}$$ in which $r\_i$ is a passage. In textual format of $R$, $r\_{i-1}$ and $r\_{i}$ is separated by a delimiter such as "[Retrieved Passage]", denoted as $d$.
>
> In Line 140 of our paper, latent variable inference is modeled using a Hidden Markov Model, where the latent concept $z$ determines the transition probability matrix between hidden states. The ``delimiter hidden state'' $ h^d $ implies that $ p(d \mid h^d, z) = 1 $, meaning that the hidden state $ h^d $ uniquely determines the output $ d $ when $ z $ is given, making the probability of observing $ d $ equal to 1.

---

> > ### Comment · Reviewer_aryh · 2024-11-30
> >
> > Q2: Can you explain why  the denominator of the first term in v(z) is always greater than $c_1$?
> > The assumption states that v(z) is greater than $c_1$ for some hidden state $h$ and not all possible hidden states. i.e. there may be hidden states for which the probability of a certain word is $p(x|h, z*)$ = 0. The summation is over all possible hidden states, so you would need a stronger assumption here than Assumption 1.

---

> ### Author Response · Authors · 2024-11-17
> **Author Response (5/5)**
>
> > **Q3.2:** How assumption 2 leads to $O(1)$ in line 765 ($p(h\_{i-1}^{d}|r\_{1:i-1},z)=O(1)$)?
>
> **A3.2:**
>  In Line 795-810 of our latest uploaded PDF, we have made this point clear.
>
> Let's decompose $p(h\_{i-1}^{d}|r\_{1:i-1},z)$ with chain rule:
> $$\begin{align}
> p(h\_{i-1}^{d}|r\_{1:i-1},z) = \sum\_h p(h\_{i-1}^{d}|h,r\_{1:i-1},z)p(h|r\_{1:i-1}z).
> \notag\end{align}$$ According to the standard assumptions of HMM [1], $h\_{i-1}^{d}$ and $r\_{1:i-1}$ are conditionally independent, so we can get:
> $$\begin{align}
> p(h\_{i-1}^{d}|r\_{1:i-1},z) = \sum\_h p(h\_{i-1}^{d}|h,z)p(h|r\_{1:i-1}z).
> \notag\end{align}$$ Assumption 2 assumes that for any delimiter hidden state $h^d$ and any other hidden state $h$, the transition probability from $h$ to $h^d$ has upper and lower bound as: $0 \leq c\_2\leq p(h^d|h,z) \leq c\_3$. Then we can get:
> $$\begin{align}
> \sum\_h p(h\_{i-1}^{d}|h,z)p(h|r\_{1:i-1}z) &= \sum\_h O(1)p(h|r\_{1:i-1}z).
> \notag \\\\
> &=O(1)\sum\_h p(h|r\_{1:i-1}z)=O(1). \notag
> \end{align}$$ Therefore, $p(h\_{i-1}^{d}|r\_{1:i-1},z)=O(1)$ in our paper can be supported by assumption 2.
>
> [1] A Tutorial on Hidden Markov Models and Selected Applications in Speech Recognition
>
> > **Q4:** It took me a while to understand how Eqn 33 could be potentially derived from Eqn 32. It would be prudent to be more rigorous here for future readers.
>
> **Our Response:**
> Thanks for your suggestion, in Line 756-783 of our latest uploaded PDF, we have made this point more rigorous for readers.  (*We found that OpenReview's rendering of latex formulas is unstable (depending on the browser version), so the display of the following formulas may be abnormal. We recommend you to read Line 756-783 of our latest uploaded paper.*)
>
> $$\begin{align}
> p(R,x\_{1:i-1}|z)=p(x\_{1:i-1}|R,z)p(R|z) \notag \text{  , // chain rule}
> \notag\end{align}$$ Eqn 26-32 shows:
> $$\begin{align}
>     v(z) &= \textrm{log}\frac{p(R,x\_{1:i-1}|z)}{p(R,x\_{1:i-1}|z^*)} \notag \text{  , // our definition in Eqn 5 of our paper.} \\\\
>     &= \textrm{log}\frac{p(x\_{1:i-1}|R,z)p(R|z)}{p(x\_{1:i-1}|R,z^*)p(R|z^*)} \notag \\\\
>     &= \textrm{log}\frac{p(x\_{1:i-1}|R,z)}{p(x\_{1:i-1}|R,z^*)} + \textrm{log}\frac{p(R|z)}{p(R|z^*)} \notag  \notag \\\\
>     & \leq  -\textrm{log}{c\_1} + \textrm{log}\frac{p(R|z)}{p(R|z^*)} . \notag
> \notag\end{align}$$ So we can get:
> $$\begin{align}
>  \textrm{log}\frac{p(x\_{1:i-1}|R,z)}{p(x\_{1:i-1}|R,z^*)} &\leq  -\textrm{log}{c\_1}\notag \\\\
> \frac{p(x\_{1:i-1}|R,z)}{p(x\_{1:i-1}|R,z^*)} &\leq \frac{1}{c\_1} \\\\
> p(x\_{1:i-1}|R,z) &\leq \frac{p(x\_{1:i-1}|R,z^*)}{c\_1}
> \notag\end{align}$$ So $p(x\_{1:i-1}|R,z)$ is bound by $O(1)$ and then we can get:
> $$\begin{align}
> p(R,x\_{1:i-1}|z) &= p(x\_{1:i-1}|R,z)p(R|z) \notag \\\\
> & \approx O(1)p(R|z) \notag \\\\
> & = \prod_{i=1}^{n} O(1)p(r\_i|r\_{1:i-1},z) \notag \text{   , // $p(R|z)=\prod\_{i=1}^{n}p(r\_i|r\_{1:i-1},z)$, $r\_i$ is the passage in $R$.}
> \notag\end{align}$$ So the Eqn 33 has been proved:
> $$\begin{align}
> p(R,x\_{1:i-1}|z) \approx \prod\_{i=1}^{n} O(1)p(r\_i|r\_{1:i-1},z).
> \notag\end{align}$$
>
> > **Q5:** Even if I assume theorem 1 is correct, in theorem 2, authors claim "approximate positive correlation", but I don't think this is a standard or well defined term. In lines 978 - 983, the authors go from upper and lower bounds to the claim on "approximate correlation" which doesn't seem theoretically sound.
>
> **Our Response:**
> Thanks for your insightful suggestion that can further improve the rigor of our paper. To solve this concern, in Line 1036-1058 of our latest uploaded PDF, we derive the maximum error for "approximation" in this approximate correlation to make it more standard. Specifically, we find that the maximum error rate $e\_{max}$ is a function of the detriment $\Upsilon$ (defined in Eqn 9 in our paper) brought by RAG, which can be described as:
> $$\begin{align}
> e\_{max} = \frac{\sqrt{\Upsilon}}{\text{exp}(\Upsilon)}.
> \notag\end{align}$$ According to Eqn 9 in our paper, the detriment $\Upsilon$ is the KL divergence between two distribution, so the value range of $\Upsilon$ is $(0,+\infty)$. So we can get:
> $$\begin{align}
> \lim_{\Upsilon \to +\infty} \frac{\sqrt{\Upsilon}}{\text{exp}(\Upsilon)} = 0.
> \notag\end{align}$$ Therefore, the greater the detriment brought by RAG , the smaller the maximum error rate of this correlation and the more robust of our theory. This shows that our method is effective to avoid the detriment brought by RAG at token-level.
>
>
> > **Q6:** Just do double check, is Eqn 29 the same as Eqn 30?
>
> **Our Response:** We are sorry for this typo, we have marked it in our latest uploaded PDF and will revise it in our final version.

---

> ### Author Response · Authors · 2024-11-30
> **Gentle Reminder**
>
> Dear Reviewer aryh
>
> Thanks for your response and your insightful suggestions. We are sorry to bother you but since the discussion phase will be ending in a few days, we would like to know can our responses address your concerns and change your assessment of our paper? If you still have any questions or concerns, we are glad to discuss with you!
>
> We sincerely appreciate your time and we are looking forward to your reply.
>
> Thank you!

---

> > ### Comment · Reviewer_aryh · 2024-11-30
> >
> > Thank you for your responses and all the updates to the paper. There were many updates that satisfied many of the weaknessess that I found. In particular, the evaluation numbers on a wider variety of tasks increase my confidence in the practical justification and all the clarifications make it easier to follow the theoretical justification. I believe the paper brings new ideas for discussion to ICLR that are justifiable purely on the basis of their practical experiments, so I'm increasing my score to 6. Having said that, I'm still not convinced with the theoretical justification.
> >
> > As a suggestion for the future, I suspect that the theoretical justification can be drastically simplified and the simpler form will have lesser errors or questionable leaps. For instance, the authors make an assumption that the retrieved passages R is a sequence of passages such that P(r_i) could depend on prior passages, and then go to great lengths (introduce delimiter tokens etc.) to show that p(r_i) is independent of many other variables. But this seems like a needless complication. Nearly always, the retrieved passages are conditionally (on the query) independent of each other. I understand the reason for starting this way, after all, the seq2seq architecture works only on concatenated strings including retrieved passages. But just because an architecture allows for more expressive interactions, doesn't mean the theoretical derivation needs to model that interaction. Perhaps it is prudent to make simplifying independence, ensure that the derivation is correct. FWIW, in the practical experiments, this is true because of the retriever setup. As a secondary exposition, one can show that the simplifying assumptions hold even in the case of concatenated passages.

---

> > > ### Author Response · Authors · 2024-11-30
> > > **Thanks for your discussion!**
> > >
> > > We are very honored that our reply can change your assessment of our paper and thank you for improving your score. As for the theoretical proof, we believe this is correct. A practical support is that our method has achieved state-of-the-art results across a diverse range of tasks and datasets, including dialogue, code generation, slot filling, language modeling, and long-form question answering. Importantly, our method is purely grounded in our theoretical framework and does not require any additional training or modules (while all baselines need). This further demonstrates the credibility and accuracy of our theory.
> > >
> > > We promise to make our theoretical proof more understandable according to your comments. Our paper takes the first step to bring essential theoretical understanding to RAG. Thank you for recognizing our contribution!

---

> ### Author Response · Authors · 2024-11-30
>
> We are willing to answer your question.
> Assumption 1 says there is some hidden state $h$ **lower bounds** $p(x|h,z*)$ > $c_1$, it means $h$ is the lower bound hidden state that makes $p(x|h,z*)$ greater than $c_1$, so all hidden states make $p(x|h,z*)$ greater than $c_1$, as you said. We apologize for this ambiguity and promise to make it clearer as you suggested.

---

### Meta-Review · Area_Chair_GkBC · 2024-12-18

**Metareview:**

This paper provides a theory to describe the trade-off between the value of external knowledge and its potential risk of misleading LLMs in next token prediction of RAG, and proposes a practical method to make pure LLM and RAG collaborate at token level. This paper connects theory with practice and has novel ideas. The reviewers raised major concerns about the presentation of the paper, and it has many writing issues, which need to be carefully addressed in the next version. One reviewer was still not convinced with the theoretical justification. After the author-reviewer discussion, all the reviewers were slightly positive about this paper. In all, this is a borderline paper and  I have mixed feelings about it, a bit positive.

**Additional Comments On Reviewer Discussion:**

Two reviewers slightly increased their scores after the author-reviewer discussion, and all the reviewers were slightly positive about this paper. However, One reviewer was still not convinced with the theoretical justification.

---

### Decision · Program_Chairs · 2025-01-22

Accept (Poster)